# TBK1-associated adapters TANK and AZI2 protect mice against TNF-induced cell death and severe autoinflammatory diseases

Andrea Ujevic [1], Daniela Knizkova [1,2], Alzbeta Synackova [1], Michaela Pribikova[1], Tijana Trivic[1,3], Anna Dalinskaya[1], Ales Drobek [2], Veronika Niederlova [2], Darina Paprckova[2,3], Roldan De Guia[4], Petr Kasparek[4], Jan Prochazka [4], Juraj Labaj[4], Olha Fedosieieva[4], Bernhard Florian Roeck [5], Ondrej Mihola[6], Zdenek Trachtulec[6], Radislav Sedlacek [4], Ondrej Stepanek [2] & Peter Draber [1,2,3] ✉

The cytokine TNF can trigger highly proinflammatory RIPK1-dependent cell death. Here, we show that the two adapter proteins, TANK and AZI2, suppress TNF-induced cell death by regulating the activation of TBK1 kinase. Mice lacking either TANK or AZI2 do not show an overt phenotype. Conversely, animals deficient in both adapters are born in a sub-Mendelian ratio and suffer from severe multi-organ inflammation, excessive antibody production, male sterility, and early mortality, which can be rescued by TNFR1 deficiency and significantly improved by expressing a kinase-dead form of RIPK1. Mechanistically, TANK and AZI2 both recruit TBK1 to the TNF receptor signaling complex, but with distinct kinetics due to interaction with different complex components. While TANK binds directly to the adapter NEMO, AZI2 is recruited later via deubiquitinase A20. In summary, our data show that TANK and AZI2 cooperatively sustain TBK1 activity during different stages of TNF receptor assembly to protect against autoinflammation.

Tumor necrosis factor (TNF) is a proinflammatory cytokine with a crucial role in mounting immune defenses. However, it is also involved in the progression of various autoimmune diseases, including rheumatoid arthritis, inflammatory bowel disease, or psoriasis. The inflammatory effects of TNF are mediated primarily by the ubiquitously expressed TNF receptor 1 (TNFR1). Stimulation of this receptor activates diverse signaling pathways, including nuclear factor-κB (NF-κB) or mitogen-activated protein kinases (MAPK), leading to the transcription of genes coding proinflammatory and anti-apoptotic proteins. However, TNFR1 stimulation can also directly trigger cell death

mediated by receptor-interacting protein kinase 1 (RIPK1)[1]. Once recruited to the membrane-bound TNF receptor signaling complex (TNF-RSC), termed complex I, RIPK1 autophosphorylates itself and translocates to the cytoplasm, forming cell death-inducing complex II[2,3]. The interaction of RIPK1 with FADD and Caspase-8 triggers apoptosis. Alternatively, RIPK1 can bind and activate kinase RIPK3, which phosphorylates MLKL, leading to its oligomerization and pore formation in the plasma membrane and subsequent necroptosis[4–8]. In order to prevent complex II formation, RIPK1 is extensively modified by polyubiquitination and phosphorylation[9–14].

[1]Laboratory of Immunity & Cell Communication, Division BIOCEV, First Faculty of Medicine, Charles University, Vestec, Czech Republic. [2]Laboratory of Adaptive Immunity, Institute of Molecular Genetics of the Czech Academy of Sciences, Prague, Czech Republic. [3]Department of Immunobiology, University of Lausanne, Epalinges, Switzerland. [4]Czech Centre for Phenogenomics and Laboratory of Transgenic Models of Diseases, Institute of Molecular Genetics of the Czech Academy of Sciences, Vestec, Czech Republic. [5]Institute for Genetics, CECAD Cluster of Excellence, University of Cologne, Cologne, Germany. [6]Laboratory of Germ Cell Development, Division BIOCEV, Institute of Molecular Genetics of the Czech Academy of Sciences, Prague, Czech Republic. ✉e-mail: peter.draber@unil.ch

Upon assembly of TNF-RSC, the adapter TRADD recruits non-degradative ubiquitin ligases TRAF2-cIAP1-cIAP2, generating K63-ubiquitin linkages, which in turn recruit linear ubiquitin chain assembly complex (LUBAC) that creates linear (M1) ubiquitin linkages[15,16]. The adapters TAB2/3 recruit kinase TAK1 to K63-ubiquitin chains, while the adapter NF-κB essential modulator (NEMO) enables the recruitment of IκB kinase α (IKKα) and IKKβ to M1 polyubiquitin linkages, leading to the activation of MAPK and NF-κB signaling pathways[17,18]. At the same time, activation of signaling responses is tightly regulated by deubiquitinases CYLD and A20[19]. Interestingly, NEMO also promotes the recruitment of TANK binding kinase 1 (TBK1), which has a dual function within TNF-RSC. On the one hand, TBK1 partially inhibits the activation of NF-κB and MAPK signaling pathways[20,21]. On the other, it phosphorylates and inhibits RIPK1 activity to prevent TNF-induced cell death[20,22]. Since the TBK1-mediated phosphorylation of RIPK1 functions as a separate cell death checkpoint from NF-κB-induced transcription of anti-apoptotic genes[23], even enhanced activation of the NF-κB pathway cannot prevent TNF-induced cell death in TBK1-deficient cells[20].

*Tbk1*[−/−] mice derived from the C57BL/6 strain die during embryonic development due to liver degeneration. This phenotype is rescued by ablation of TNFR1 or inactivation of RIPK1 kinase activity, indicating that it is driven by aberrant TNF-induced cell death[20,24]. Similarly, mice harboring enzymatically inactive TBK1 suffer from embryonic lethality caused by increased RIPK1-induced cell death[25]. *Tbk1*[−/−] mice on the 129S5 background are viable, but they are born at decreased Mendelian frequency and suffer from autoinflammation and immune infiltrates in multiple organs[26]. Human patients lacking TBK1 developed a severe autoimmune disease that could be treated by blocking TNF[27]. These findings collectively demonstrate the critical role of TBK1 in protecting against TNF-induced inflammation and cell death. TBK1 is also a potential target for cancer treatment, as its inhibition significantly potentiates anti-PD-L1 immunotherapy due to the enhanced apoptosis and necroptosis of tumor cells triggered by TNF and IFNγ[28].

TBK1 is a pleiotropic kinase that modulates various processes, including metabolism, autophagy, interferon signaling, and cell death[29]. The broad range of TBK1 functions stems from its ability to interact with different adapters that connect it with distinct signaling complexes[30,31]. Furthermore, TBK1 can directly bind to immune receptors, such as the stimulator of interferon genes (STING) that detects cytosolic DNA derived from intracellular pathogens[32]. In the context of the TNF-RSC, TBK1 was proposed to be recruited via adapters TRAF family member associated NF-κB activator (TANK) and 5-azacytidine induced 2 (AZI2, also known as NAP1)[22]. However, the role of these two adapters in protecting against TNF-induced cell death has been enigmatic. *Tank*[−/−] mice are born at the expected Mendelian ratio and appear normal, although they develop glomerulonephritis over time, which is mediated by aberrant MyD88-signaling[33]. *Azi2*[−/−] mice show no overt phenotype[34].

Here, we analyze whether TANK and AZI2 are partially redundant in protecting against TNF-driven autoinflammation. We show that combined deletion of TANK and AZI2 strongly suppresses TNF-induced activation of TBK1. This is associated with a modest increase in NF-κB signaling and a significant enhancement of TNF-induced cell death in TANK/AZI2-deficient cells. Consistently, mice lacking both proteins develop severe TNFR1-driven autoinflammatory illness. Altogether, TANK and AZI2 are crucial components of the TNF-RSC, protecting against aberrant TNF-induced signaling and cell death.

## Results

### TANK and AZI2 protect against TNF-induced cell death

To investigate whether adapters TANK and AZI2 contribute to TNF-induced activation of TBK1, we generated murine stromal ST2 cell lines lacking TANK, AZI2, or both proteins using CRISPR/Cas9. We stimulated these cells with Strep-Flag tagged TNF (SF-TNF) for 15 minutes and isolated the TNF-RSC through Flag immunoprecipitation from

cellular lysates. Compared to wild-type (WT) cells, the recruitment and phosphorylation of TBK1 within the TNF-RSC were reduced in *Tank* knockout (KO) and abolished in *Tank/Azi2* double KO (DKO) cells (Fig. 1a, b).

Subsequently, we analyzed the role of the two adapters in TNF-induced activation of signaling pathways. We observed that TBK1 activating phosphorylation at serine 172[35,36] was reduced in *Tank*[KO] cells compared to wild-type (WT) or *Azi2*[KO] cells and strongly suppressed in *Tank/Azi2*[DKO] cells (Fig. 1c, d). In contrast, the induction of the NF-κB pathway, detected as activating phosphorylation of IKKα/β, was slightly but very reproducibly increased in cells lacking both adapters (Fig. 1c, d). Similarly, a detailed time-course analysis of TNF-induced signaling revealed that while TBK1 phosphorylation was suppressed, IKKα/β activating phosphorylation was both enhanced and prolonged in *Tank/Azi2*[DKO] cells compared to WT cells, which also correlated with prolonged phosphorylation of its downstream target p105 (Supplementary Fig. 1a, b). Activated IKKα/β phosphorylates IκBα, which is rapidly degraded[37]. Indeed, IκBα was barely detectable 15 minutes after TNF stimulation in both WT and *Tank/Azi2*[DKO] cells (Fig. 1c, d and Supplementary Fig. 1a, b). However, *Tank/Azi2*[DKO] cells had a markedly prolonged phase of IκBα phosphorylation and degradation detected at 30 minutes upon stimulation (Fig. 1c, d and Supplementary Fig. 1a, c), which correlated with enhanced and prolonged phosphorylation of IKKα/β. Altogether, TANK and AZI2 adapters are required for TBK1 activating phosphorylation, while they partially suppress IKKα/β activation and the induction of NF-κB pathway. These results align with previous studies showing enhanced TNF-induced activation of IKKα/β in TBK1-deficient cells[20,21].

TBK1 is not required to prevent spontaneous activation of RIPK1 and induction of cell death in unstimulated cells. However, it inhibits the autophosphorylation of RIPK1 on Serine 166 and subsequent proteolytic activation of Caspase-8 upon TNF stimulation[20,22]. TNF-induced Caspase-8 cleavage was enhanced in *Tank/Azi2*[DKO] cells compared to WT cells (Fig. 1e, f). When cells were treated with TNF in the presence of the caspase inhibitor zVAD, which stabilizes complex II formation, we observed robust RIPK1 autophosphorylation in *Tank/Azi2*[DKO] cells (Fig. 1e, f). In accord, real-time cell imaging revealed that *Tank/Azi2*[DKO] cells were susceptible to TNF-induced cell death, which was partially inhibited by zVAD or the RIPK1 inhibitor Necrostatin-1s (Nec-1s) and rescued by simultaneous treatment with both inhibitors (Fig. 1g, h and Supplementary Fig. 1d). Similarly, the detection of dead cells by propidium iodide (PI) uptake via flow cytometry revealed that *Tank/Azi2*[DKO] cells are sensitive to TNF induced cell death (Supplementary Fig. 1e). This method detected even higher proportion of TNF-induced cell death in *Tank/Azi2*[DKO] cells, which likely reflects differences in sample preparation.

Regulation of RIPK1 activity and NF-κB-induced transcription of anti-apoptotic genes are two separate cell death checkpoints. Blocking NF-κB activation with IKKα/β inhibitor TPCA-1 further potentiated TNF-induced cell death in *Tank/Azi2*[DKO] ST2 cells (Supplementary Fig. 1f), indicating that IKKα/β-mediated cell death checkpoint is operating in *Tank/Azi2*[DKO] cells. We also observed strongly suppressed phosphorylation of TBK1, slightly enhanced phosphorylation of IKKα/β, and increased cell death upon TNF stimulation in human HeLa *TANK/AZI2*[DKO] cells (Supplementary Fig. 1g–j). Combined, these data suggest that TANK and AZI2 regulate TNF-induced signaling and cell death by enabling TBK1 recruitment to TNF-RSC.

### TANK and AZI2 protect against autoinflammatory disease in mice

To investigate the potential redundant function of the adapters TANK and AZI2 in vivo, we generated *Tank*[−/−] and *Azi2*[−/−] mice using the CRISPR/Cas9 approach (Fig. 2a, b). As expected, *Tank*[−/−] and *Azi2*[−/−] mice were born in normal ratios and were indistinguishable from their WT littermates (Fig. 2c, d). In contrast, upon the breeding of *Azi2*[−/−]/*Tank*[+/−]

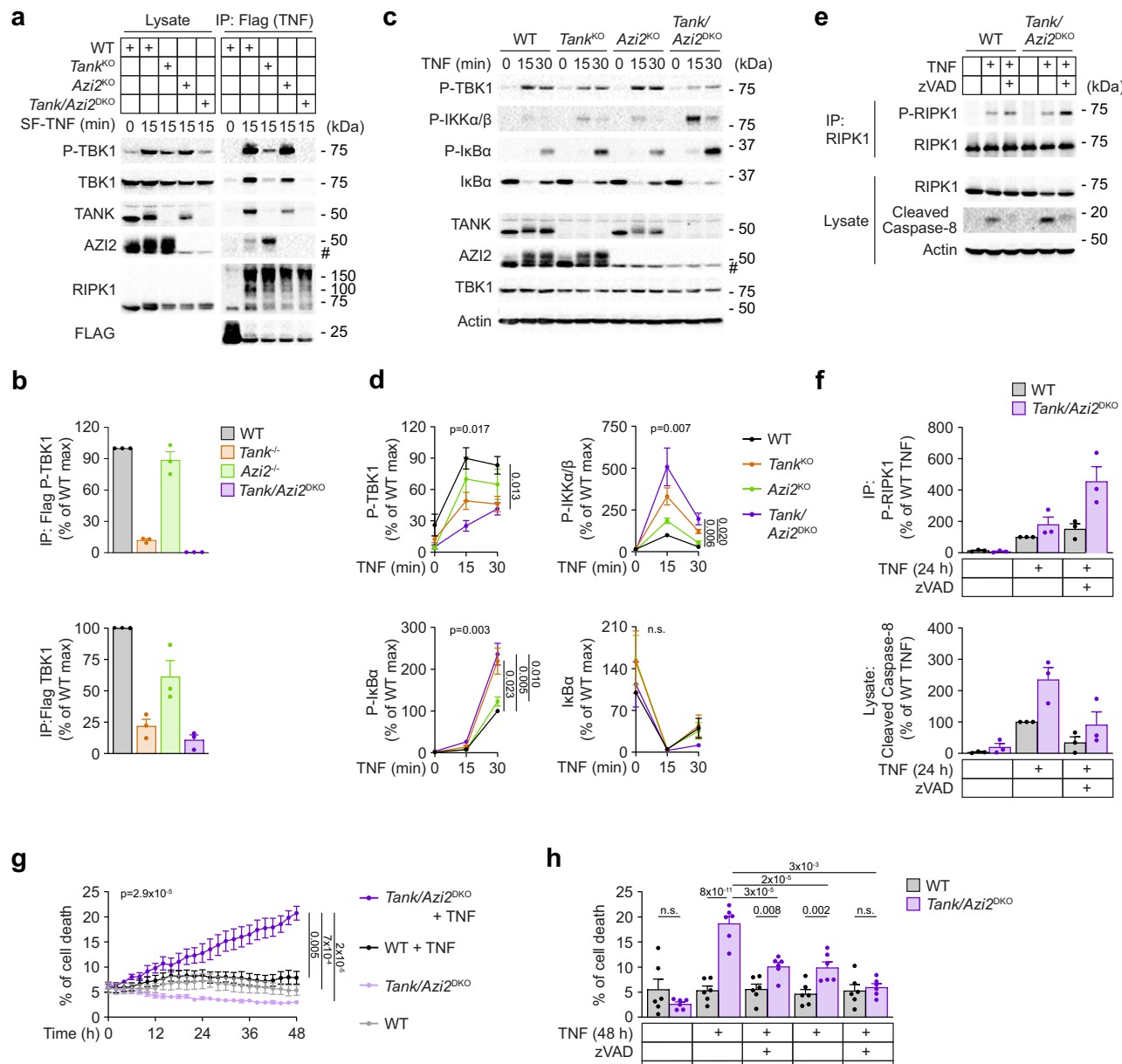

**Fig. 1 | Adapters TANK and AZI2 are required for TNF-induced TBK1 activation.**
**a**, **b** Immunoblot analysis of TNF-RSC isolated from ST2 WT, *Tank*^KO, *Azi2*^KO, or *Tank/Azi2*^DKO cell lines. Cells were stimulated with SF-TNF (500 ng/ml) as indicated, and lysates were subjected to anti-Flag immunoprecipitation. As a control, SF-TNF was added post-lysis to unstimulated samples. A representative experiment (**a**) and quantification of recruitment of indicated proteins to TNF-RSC from three independent experiments normalized to WT cells, mean + SEM (**b**). **c**, **d** Immunoblot analysis of lysates of ST2 WT, *Tank*^KO, *Azi2*^KO, or *Tank/Azi2*^DKO cells stimulated with TNF (100 ng/ml) as indicated. A representative experiment (**c**) and quantification of three independent experiments normalized to the maximal response of WT cells, mean ± SEM (**d**). Statistical analysis was based on the area under the curve (AUC) for particular cell lines in each experiment, one-way ANOVA (*p*-value shown) with Tukey´s post-tests. **e**, **f** Immunoblot analysis of lysates from ST2 WT or *Tank/Azi2*^DKO cells that were unstimulated or stimulated for 24 h with TNF (250 ng/ml) and zVAD (20 μM) as indicated and subjected to RIPK1 immunoprecipitation. A representative experiment (**e**) and quantification of phosphorylated RIPK1 and cleaved Caspase-8 from three independent experiments normalized to WT cells stimulated with TNF alone, mean + SEM (**f**). **g** ST2 WT and *Tank/Azi2*^DKO cells were unstimulated or stimulated with TNF (100 ng/ml), and the induction of cell death was monitored every 2 h using the Incucyte imaging system. The mean ± SEM from six separate experiments using two different *Tank/Azi2*^DKO ST2 clones is shown. Statistical analysis was based on the AUC for different cell lines and treatments in each experiment, one-way ANOVA (*p*-value shown) with Tukey´s post-tests. **h** ST2 WT and *Tank/Azi2*^DKO cells were unstimulated or stimulated with TNF (100 ng/ml) alone or in the presence of zVAD (20 μM), Nec-1s (20 μM), or both inhibitors. Induction of cell death after 48 h was measured using the Incucyte imaging system. The mean + SEM from six independent experiments using two different *Tank/Azi2*^DKO ST2 clones is shown, two-way ANOVA with Tukey's post-tests. n.s., not significant. #, nonspecific band.

mice, we observed that the *Tank/Azi2*^DKO animals were born at a markedly sub-Mendelian ratio, indicating partial embryonic lethality (Fig. 2c). *Tank/Azi2*^DKO pups exhibited delayed hair growth, and by four weeks of age, they completely lacked body hair except for the head and tail base regions (Fig. 2d). Their fur eventually grew, but the mice developed severe dermatitis within a few months (Fig. 2e). The median

survival of *Tank/Azi2*^DKO animals was only 25 weeks (Fig. 2f). These findings were confirmed in animal strains derived from independent founders harboring different mutations in *Tank* and *Azi2* alleles. In this case, we crossed *Tank*^−/−/*Azi2*^+/− mice to obtain *Tank/Azi2*^DKO animals, which were again born at a low ratio and displayed delayed hair growth and short lifespan (Supplementary Fig. 2a–e).

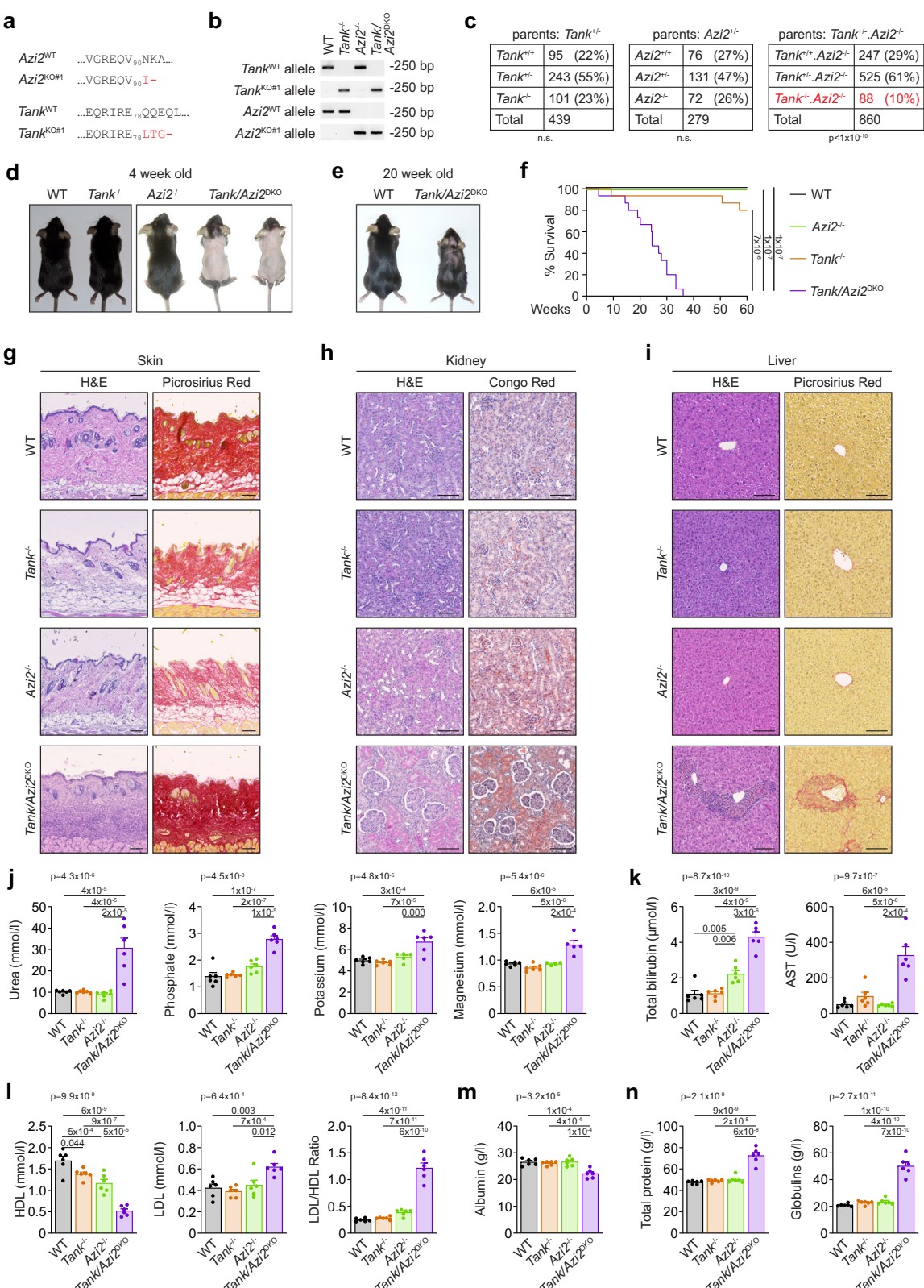

**Fig. 2 | *Tank/Azi2*^DKO mice develop severe autoinflammation. a, b** Translations of DNA sequences (**a**) and an example of routine PCR genotyping (**b**) of the indicated *Tank* and *Azi2* alleles. **c** Genotype and counts of pups born to parents of the indicated genotypes, two-tailed chi-square test. **d, e** Representative photographs of mice of the indicated genotype at the age of 4 weeks (**d**) and 20 weeks (**e**). **f** Survival curves of mice of the indicated genotype (*n* = 15 per group), log-rank Mantel-Cox test. **g–i** Tissue sections stained with H&E or the indicated dye. Skin (**g**), kidney (**h**),

or liver (**i**) samples were collected from 20-24-week-old mice of the indicated genotype. Representative images are based on the analysis of five mice per group. **j–n** The concentration of different biochemical markers in the blood plasma collected from 20-24-week-old mice of the indicated genotype (*n* = 6 per group), mean + SEM, one-way ANOVA (*p*-value is shown) with Tukey's post-tests. n.s., not significant. The scale bar is 100 μm.

To elucidate the likely cause of death of *Tank/Azi2*^DKO animals, we analyzed organs isolated from 20-24-week-old mice by histology. No apparent differences were observed between WT, *Tank*^-/- and *Azi2*^-/- mice at this age. In contrast, histological examination of *Tank/Azi2*^DKO mice tissue samples revealed a severe autoinflammatory disease affecting multiple organs. Hematoxylin and eosin (H&E) staining of skin sections from *Tank/Azi2*^DKO mice revealed extensive immune cell infiltrations. The subdermal fat was completely missing, which was also demonstrated by staining with picrosirius red that binds to collagen fibers (Fig. 2g and Supplementary Fig. 2f). Kidney sections stained with H&E displayed glomerulonephritis characterized by swollen glomeruli, and Congo red staining identified massive amyloid deposits in the kidneys (Fig. 2h and Supplementary Fig. 2g). H&E staining of liver sections revealed numerous granulomas and picrosirius red staining showed collagen-filled scars and bile duct fibrosis (Fig. 2i and Supplementary Fig. 2h). The tissue damage in the skin, kidney, and liver in *Tank/Azi2*^DKO animals was accompanied by enhanced apoptosis detected by TUNEL staining (Supplementary Fig. 2i). Additionally, we observed massive immune infiltrations in the pancreas, skeletal muscles, and white adipose tissues (Supplementary Fig. 2j).

The internal organ damage observed in *Tank/Azi2*^DKO mice was further confirmed by biochemical analysis of blood plasma. *Tank/Azi2*^DKO mice exhibited significantly elevated urea, phosphate, potassium, and magnesium levels, indicating severe kidney damage, which was not observed in WT, *Tank*^-/- or *Azi2*^-/- mice (Fig. 2j). Moreover, we detected substantial increases in serum levels of total bilirubin and aspartate aminotransferase (AST) in *Tank/Azi2*^DKO mice compared to WT animals or mice lacking either adapter alone, indicating liver damage (Fig. 2k). In terms of lipid metabolism, *Tank/Azi2*^DKO mice exhibited a strongly decreased level of plasma cholesterol concentration, while the concentration of triglycerides remained unchanged in *Tank/Azi2*^DKO mice compared to WT animals (Supplementary Fig. 2k). High-density lipoprotein (HDL) concentration was lowered in *Tank*^-/-, or *Azi2*^-/- animals and further significantly decreased in *Tank/Azi2*^DKO mice as compared to WT mice (Fig. 2l). In contrast, the plasma concentration of low-density lipoprotein (LDL), which contributes to cholesterol build-up in arteries, was substantially elevated in *Tank/Azi2*^DKO mice. This was reflected by a very high ratio of LDL to HDL in *Tank/Azi2*^DKO mice (Fig. 2l), indicating an increased risk for cardiovascular failure.

Plasma albumin levels were decreased in *Tank/Azi2*^DKO mice (Fig. 2m), which is consistent with ongoing autoinflammatory disease and liver malfunction. These animals exhibited increased concentrations of total plasma proteins, suggesting an elevation in serum globulins and potentially an imbalance of the immune system and antibody production (Fig. 2n). In conclusion, the shortened lifespan of *Tank/Azi2*^DKO animals seems to be caused by multiorgan inflammation. These findings provide strong evidence that TANK and AZI2 can compensate for each other in vivo to protect against severe autoinflammatory disease.

### *Tank/Azi2*^DKO mice have dysregulated antibody production

The increased concentration of globulins in *Tank/Azi2*^DKO mice prompted us to analyze the concentration of different antibody isotypes in plasma. *Azi2*^-/- mice showed similar levels of plasma antibodies as WT animals. In contrast, *Tank*^-/- mice displayed increased concentrations of all antibody isotypes, as reported previously[33]. *Tank/Azi2*^DKO mice had similar plasma concentrations of IgM, IgG1, IgG2a, IgG2b, and IgE antibodies to *Tank*^-/- animals, while IgA antibody levels were substantially increased (Fig. 3a).

Spleens from *Tank/Azi2*^DKO mice were enlarged, especially when normalized to body weight (Fig. 3b). Their histological analysis by H&E staining revealed disrupted splenic architecture with poorly distinguishable boundaries between white and red pulp (Fig. 3c). Flow cytometry analysis of *Tank/Azi2*^DKO spleen demonstrated increased cell

counts with normal splenocytes viability (Fig. 3d). We detected massive splenic infiltration of macrophages and neutrophils, while the proportion of dendritic cells (DC) was decreased (Fig. 3e). The ratio of splenic B cells was similar in WT, *Tank*^-/-, *Azi2*^-/-, and *Tank/Azi2*^DKO animals (Fig. 3f). Upon activation, mature IgD^+ IgM^Low B cells undergo isotype switching to produce antibodies of different isotypes. *Tank/Azi2*^DKO mice exhibited a substantial increase in the proportion of isotype-switched B cells, characterized as CD19^+ IgM^- IgD^-, which corresponds to the high concentration of antibodies observed in their plasma (Fig. 3g). Similarly, a highly increased percentage of isotype-switched B cells was observed in the lymph nodes (Fig. 3h, i).

To elucidate whether the increased proportion of isotype-switched B cells is driven by ongoing autoinflammatory disease or is a cell-intrinsic process, we performed competitive bone-marrow transfer. Bone marrow cells from WT and *Tank/Azi2*^DKO mice were mixed in a 1:1 ratio and transplanted into lethally irradiated mice. The emergence of immune cells derived from *Tank/Azi2*^DKO progenitors was significantly suppressed compared to cells derived from WT progenitors (Supplementary Fig. 3a, b). However, the percentage of isotype-switched B cells was substantially enhanced in *Tank/Azi2*^DKO compared to WT B cells in both spleen and lymph nodes (Supplementary Fig. 3c–f). Altogether, our results indicate that TANK and AZI2 function as crucial but partially redundant regulators of hematopoiesis, isotype switching, and antibody production by B cells.

### Enhanced signaling and cell death in cells isolated from *Tank/Azi2*^DKO mice

Mouse embryonic fibroblasts (MEFs) isolated from *Tank*^-/- or *Azi2*^-/- animals had decreased phosphorylation of TBK1, which was severely impaired in *Tank/Azi2*^DKO compared to WT cells (Supplementary Fig. 4a, b). Similarly to *Tbk1*^-/- MEFs[20], *Tank/Azi2*^DKO MEFs had slightly increased activation of IKKα/β, JNK, and p38 MAPK signaling pathways (Supplementary Fig. 4c, d). Moreover, the recruitment and phosphorylation of TBK1 within TNF-RSC were strongly reduced in *Tank/Azi2*^DKO MEFs compared to WT or *Tank*^-/- MEFs, and this correlated with enhanced NEMO recruitment (Supplementary Fig. 4e, f).

TNF stimulation of *Tank/Azi2*^DKO bone marrow-derived macrophages (BMDMs) led to strongly suppressed phosphorylation of TBK1, which was accompanied by slightly enhanced activation of NF-κB and JNK MAPK signaling pathways (Fig. 4a, b). The stimulation of BMDMs with TNF triggers the expression of TNF-induced genes that can be identified by RNA sequencing (RNAseq) (Supplementary Fig. 4g and Supplementary Dataset 1). *Tank/Azi2*^DKO BMDMs had enhanced basal expression of TNF-responsive genes, including transcription of anti-apoptotic genes *Traf1*, *Cflar*, *Birc2*, and *Birc3*[38,39], which likely reflects the increased basal expression of TNF in these cells. The transcription of TNF-induced genes was further enhanced in *Tank/Azi2*^DKO compared to WT BMDMs upon TNF stimulation (Fig. 4c, d and Supplementary Fig. 4h). In contrast, we did not observe increased transcription of anti-apoptotic Bcl2 family members after two hours of TNF stimulation (Supplementary Fig. 4h).

*Tank/Azi2*^DKO BMDMs had increased basal level of cell death, and TNF stimulation of these cells showed enhanced induction of cell death in comparison to WT cells (Fig. 4e). Altogether, these data demonstrate that increased activation of the NF-κB pathway and induction of anti-apoptotic genes in *Tank/Azi2*^DKO cells is not sufficient to prevent cell death caused by a block in TBK1 recruitment to TNF-RSC, which is required to control the activation of RIPK1.

### TNFR1 is the primary driver of autoinflammation in *Tank/Azi2*^DKO mice

Since TBK1 activation is necessary to protect against TNF-induced lethality in vivo[22], we aimed to investigate the role of TANK and AZI2 in safeguarding against a sublethal dose of TNF administered via tail-vein injection. While WT, *Tank*^-/-, and *Azi2*^-/- mice tolerated the

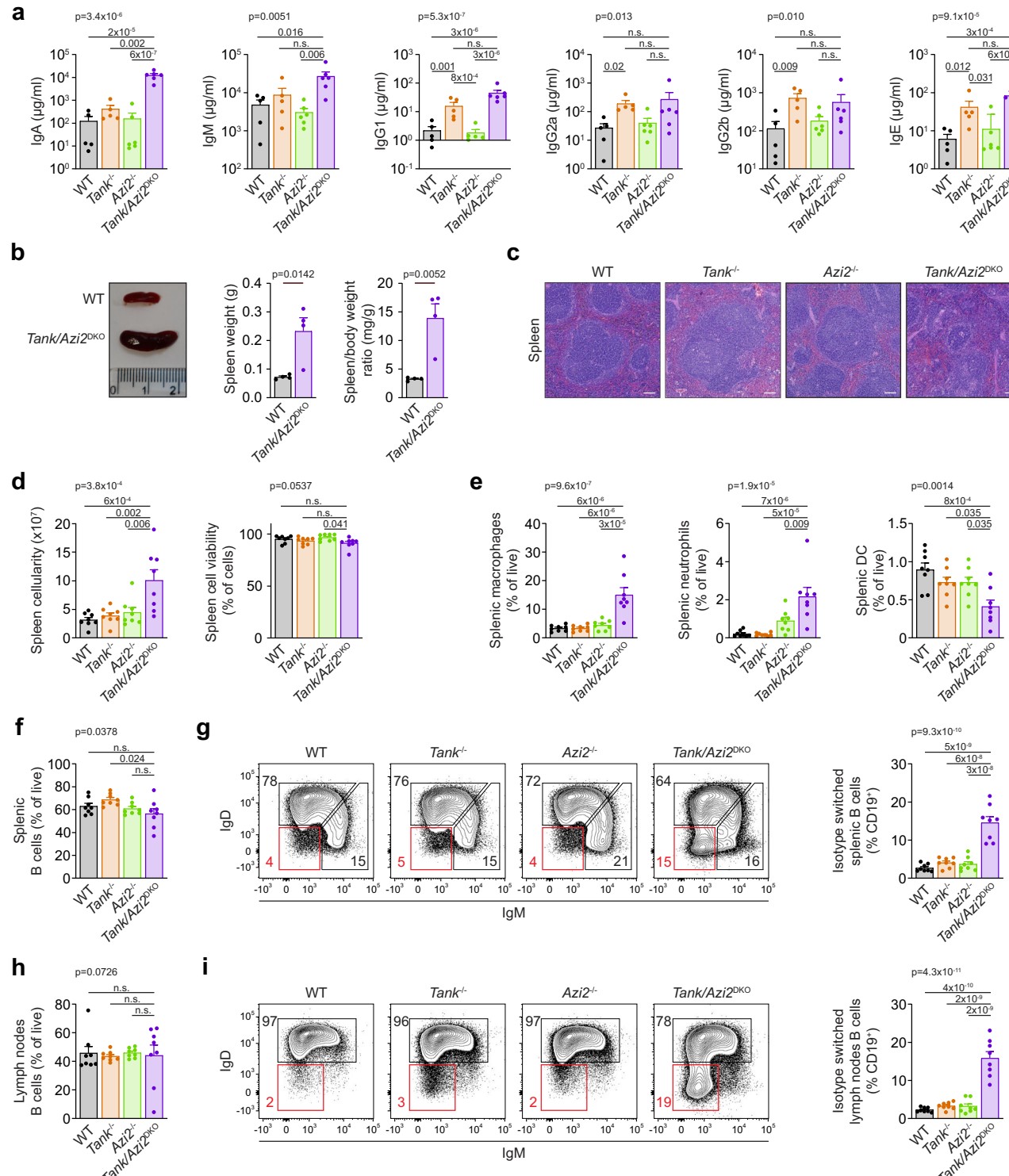

**Fig. 3 | *Tank/Azi2*^DKO mice have dysregulated antibody production. a** Titers of different antibody isotypes in the blood plasma from 20-24-week-old mice of the indicated genotype (n = 5 for WT and *Azi2*^−/−, 6 for *Tank*^−/− and *Tank/Azi2*^DKO), mean + SEM, one-way ANOVA (*p*-value is indicated) with Tukey's post-tests. **b** Representative photograph and weight of spleens isolated from 8-10-week-old WT and *Tank/Azi2*^DKO mice (*n* = 4 per group), mean + SEM, unpaired two-tailed t-test. **c** H&E staining of spleen tissue sections from 20-24-week-old mice of the indicated genotype. Representative images are based on analyzing samples from five mice per group. **d, e** Flow cytometry measurement of cellularity and viability of splenocytes from 8-12-week-old mice of the indicated genotype (**d**), followed by the analysis of splenic myeloid cells (CD3^−, CD19^−, NK1.1^−, CD11b^+) separated into macrophages (CD11c^−, Ly6G^−), neutrophils (CD11c^−, Ly6G^+), and dendritic cells (DC) (CD11c^+) (**e**) (*n* = 8 per group), mean + SEM, one-way ANOVA (*p*-value is indicated) with Tukey's post-tests. **f–i** Flow cytometry analysis of B cells from the spleen (**f**–**g**) and lymph nodes (**h**, **i**) from 8-12-week-old mice (*n* = 8 per group). The proportion of B cells (CD19^+) (**f**, **h**) and the percentage of isotype-switched B cells (IgM^−, IgD^−) in organs isolated from the indicated mouse strains, mean + SEM, one-way ANOVA (*p*-value is indicated) with Tukey's post-tests (**g**, **i**). n.s., not significant. The scale bar is 100 µm.

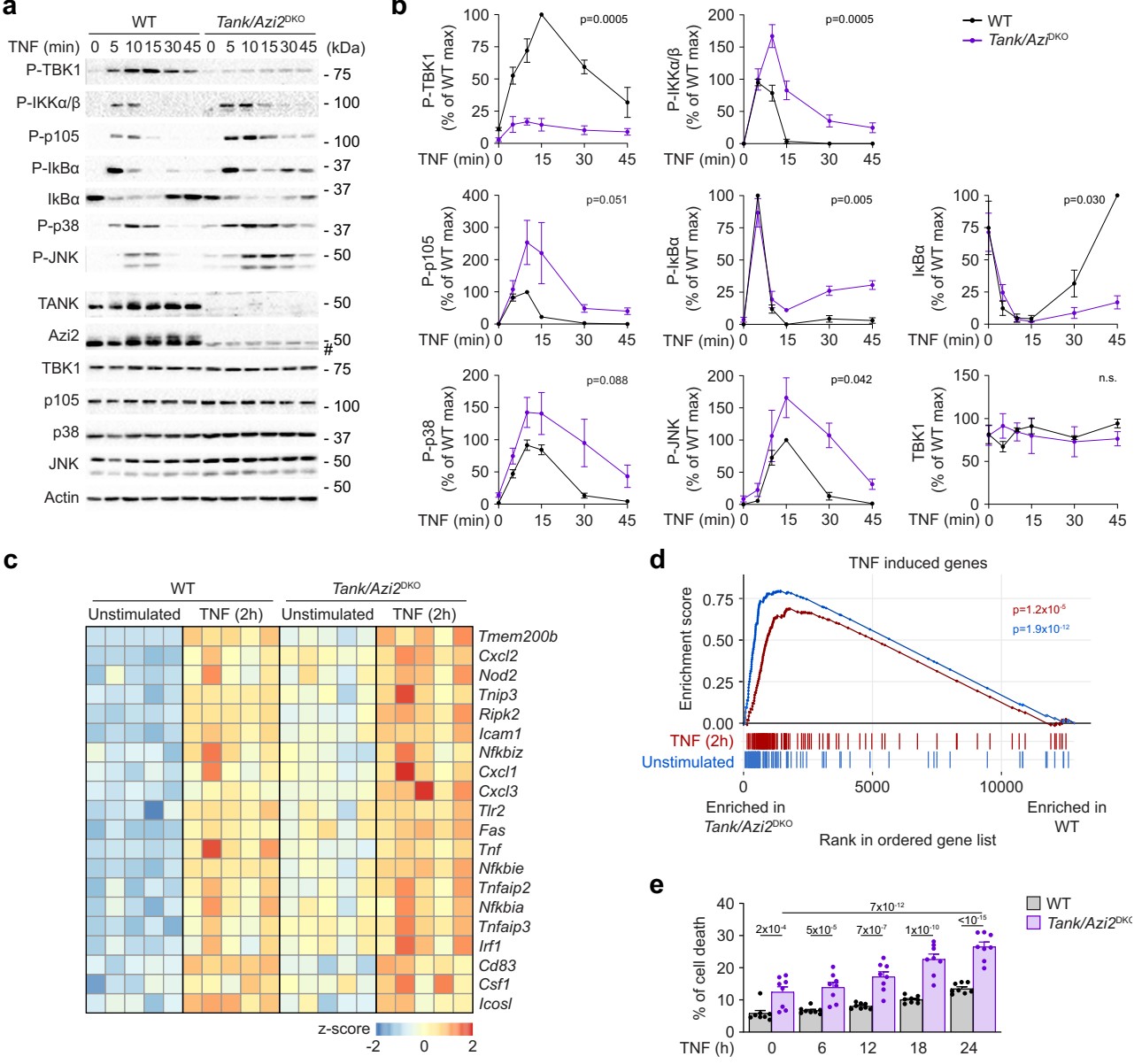

**Fig. 4 | TNF-induced TBK1 activation is impaired in *Tank/Azi2*ᴰᴷᴼ macrophages.**
**a**, **b** Immunoblot analysis of lysates of bone marrow-derived macrophages (BMDMs) isolated from WT or *Tank/Azi2*ᴰᴷᴼ mice and stimulated with TNF (100 ng/ml) as indicated. A representative experiment (**a**) and quantification of three independent experiments using BMDMs independently isolated from 3 animal pairs. Data were normalized to the maximal response of WT cells, mean ± SEM (**b**). Statistical analysis was based on the AUC for each cell line, unpaired two-tailed t-test. **c**, **d** RNAseq analysis of WT and *Tank/Azi2*ᴰᴷᴼ BMDMs stimulated for two hours with TNF (100 ng/ml). The BMDMs were independently isolated from 3 pairs of animals, and RNA was isolated in five separate experiments. Heatmap shows the expression of selected TNF-responsive genes (**c**). Gene set enrichment analysis

(GSEA) shows the enrichment of 105 genes induced by TNF stimulation (log₂ fold change >1 and padj < 0.01; samples from WT and *Tank/Azi2*ᴰᴷᴼ BMDMs were analyzed together). The ranked gene list represents the contrast between unstimulated *Tank/Azi2*ᴰᴷᴼ and WT BMDMs (blue) or stimulated *Tank/Azi2*ᴰᴷᴼ and WT BMDMs (red). The *p*-value was estimated using an adaptive multi-level split Monte-Carlo scheme in the fgsea R package (**d**). **e** Analysis of cell death induced upon TNF (100 ng/ml) stimulation of WT or *Tank/Azi2*ᴰᴷᴼ BMDMs. Induction of cell death was detected via the Incucyte imaging system. Mean + SEM from eight separate experiments using samples isolated from 4 animal pairs, two-way ANOVA with Tukey's post-tests. n.s., not significant. #, nonspecific band.

injection well, *Tank/Azi2*ᴰᴷᴼ animals rapidly succumbed to the treatment (Supplementary Fig. 5a). Intriguingly, the analysis of plasma of 20-24-week-old *Tank/Azi2*ᴰᴷᴼ mice, which suffer from autoinflammation, revealed significantly elevated plasma levels of proinflammatory cytokines TNF, IFNγ, IL-1β, and IL-17, compared to WT mice (Supplementary Fig. 5b). The increased susceptibility of *Tank/Azi2*ᴰᴷᴼ mice to TNF-induced lethality, coupled with the increased levels of plasma TNF, prompted us to explore whether the autoinflammatory disease observed in *Tank/Azi2*ᴰᴷᴼ mice is driven by aberrant TNFR1 signaling.

We crossed *Tank/Azi2*ᴰᴷᴼ animals with a mouse strain deficient in TNFR1 (*Tnfrsf1a*⁻ᐟ⁻) to obtain *Tank/Azi2/Tnfrsf1a* triple KO (TKO) mice. These animals were born at a nearly Mendelian ratio, indicating a substantial rescue of the embryonic lethality phenotype (Fig. 5a). *Tank/Azi2/Tnfrsf1a*ᵀᴷᴼ mice did not experience the delayed hair growth observed at four weeks of age in *Tank/Azi2*ᴰᴷᴼ animals (Fig. 5b) and did not develop dermatitis by 30 weeks of age (Fig. 5c). Importantly, the survival of *Tank/Azi2/Tnfrsf1a*ᵀᴷᴼ mice was significantly improved compared to *Tank/Azi2*ᴰᴷᴼ mice, indicating a substantial rescue of the phenotype (Fig. 5d).

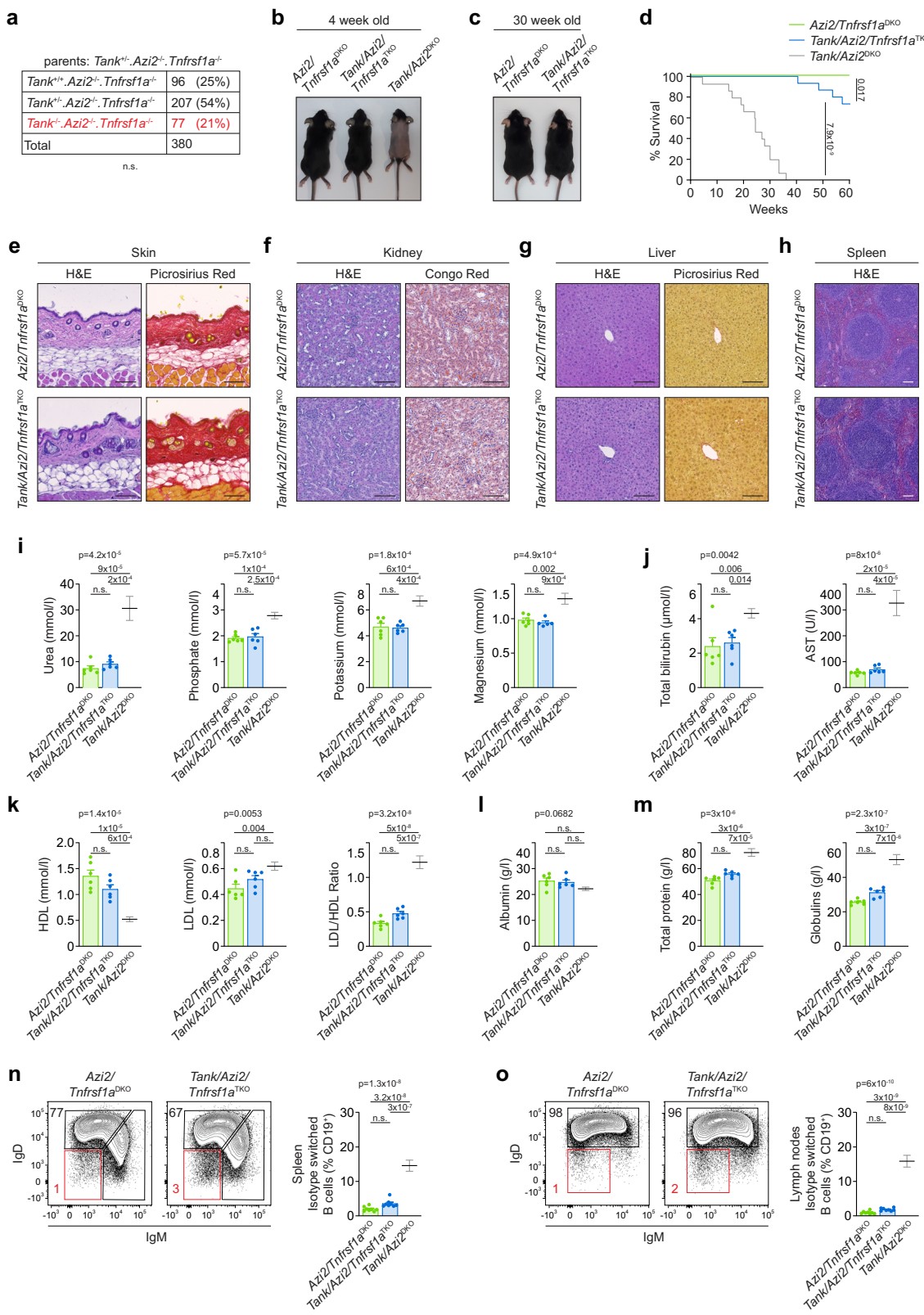

Histological analysis of organs isolated from five months old *Tank/Azi2/Tnfrsf1a*TKO mice demonstrated that they did not suffer from autoinflammation and appeared similar to control *Azi2/Tnfrsf1a*DKO animals. Notably, *Tank/Azi2/Tnfrsf1a*TKO mice did not exhibit aberrant immune cell infiltration in the skin and had normal subdermal fat deposition (Fig. 5e and Supplementary Fig. 5c). Furthermore, they did not show signs of glomerulonephritis or amyloid deposits in the kidneys, and the liver was devoid of granulomas and collagenous scars (Fig. 5f, g and Supplementary Fig. 5d, e). Additionally, no immune infiltrations were detected in the pancreas, skeletal muscle, or white adipose tissue samples isolated from these animals (Supplementary Fig. 5f). Finally, *Tank/Azi2/Tnfrsf1a*TKO mice displayed standard splenic architecture (Fig. 5h).

**Fig. 5 | TNFR1 is the primary driver of autoinflammation in *Tank/Azi2*^DKO^ mice. a** Genotype and counts of pups born to *Tank*^+/−^/*Azi2*^−/−^/*Tnfrsf1a*^−/−^ parents, two-tailed chi-square test. (**b, c**) Representative photographs of mice of the indicated genotype at the age of 4 weeks (**b**) and 30 weeks (**c**). **d** Survival curves of mice of the indicated genotype (*n* = 15 per group). The comparison with *Tank/Azi2*^DKO^ mice (based on Fig. 2) is shown in gray, log-rank Mantel-Cox test. **e**−**h** Tissue sections stained with H&E or the indicated dye. Skin (**e**), kidney (**f**), liver (**g**), or spleen (**h**) samples were collected from 20-24-week-old mice of the indicated genotype. Representative images are based on the analysis of five mice per group. **i**−**m** The concentration of different biochemical markers in the blood plasma isolated from 20-24-week-old *Azi2/Tnfrsf1a* ^DKO^ or *Tank/Azi2/Tnfrsf1a* ^TKO^ mice (*n* = 6 per group). The comparison with *Tank/Azi2*^DKO^ mice (based on Fig. 2) is shown in gray, mean + SEM, one-way ANOVA (*p*-value is indicated) with Tukey's post-tests. **n, o** Flow cytometry analysis of the proportion of isotype-switched B cells (CD19^+^, IgM^−^, IgD^−^) in the spleen (**n**) and lymph nodes (**o**) isolated from 8-12-week-old mice of the indicated genotype (*n* = 8 per group). The comparison with *Tank/Azi2*^DKO^ mice (based on Fig. 3) is shown in gray, mean + SEM, one-way ANOVA (*p*-value is indicated) with Tukey's post-tests. n.s., not significant. The scale bar is 100 μm.

Further evidence supporting the rescue of *Tank/Azi2*^DKO^ mice by TNFR1 ablation was obtained through plasma biochemical analysis. Normal plasma urea, phosphate, potassium, and magnesium levels indicated restored kidney function in *Tank/Azi2/Tnfrsf1a*^TKO^ animals (Fig. 5i). Similarly, total plasma bilirubin and AST were at normal levels in these mice, indicating rescued liver function (Fig. 5j). Regarding lipid metabolism, the *Tank/Azi2/Tnfrsf1a*^TKO^ mice exhibited comparable total cholesterol levels as control mice (Supplementary Fig. 5g). The concentration of HDL, LDL, and the ratio between LDL and HDL were rescued in *Tank/Azi/Tnfrsf1a*^TKO^ mice (Fig. 5k). Finally, we observed no significant decrease in plasma albumin or increase in total protein concentration in *Tank/Azi2/Tnfrsf1a*^TKO^ mice compared to control animals, indicating no significant increase in plasma globulins (Fig. 5l, m). *Tank/Azi2/Tnfrsf1a*^TKO^ mice did not show significantly increased number of splenocytes compared to control animals (Supplementary Fig. 5h) and had a normal proportion of B cells in spleen, albeit it was mildly increased in the lymph nodes (Supplementary Fig. 5i). Importantly, *Tank/Azi2/Tnfrsf1a*^TKO^ did not have significantly elevated proportion of isotype-switched B cells (Fig. 5n, o). These findings align with a previous report showing impaired isotype switching in B cells from *Tnfrsf1a*^−/−^ mice[40].

Interestingly, while *Tank/Azi2*^DKO^ males could not produce offspring, *Tank/Azi2/Tnfrsf1a*^TKO^ males were fertile (Supplementary Fig. 5j). Histological analysis of the testes revealed that seminiferous tubules of the *Tank/Azi2*^DKO^ mice harbored spermatozoa (Supplementary Fig. 5k). However, their testes weight was decreased, and sperm count, sperm motility, and progressive sperm motility were substantially reduced compared to control *Azi2*^−/−^ and *Tank/Azi2/Tnfrsf1a*^TKO^ mice (Supplementary Fig. 5l, m). These results show that the autoinflammatory disease and male sterility observed in *Tank/Azi2*^DKO^ mice are primarily driven by TNFR1 signaling.

## RIPK1 activity partially mediates the phenotype of *Tank/Azi2*^DKO^ mice

Given the crucial role of RIPK1 as a mediator of TNF-induced cell death[41], we aimed to investigate whether the expression of the kinase-dead form of RIPK1 (RIPK1^KD^) with the inactivating mutation D138A could rescue *Tank/Azi2*^DKO^ mice. Introducing the inactivating mutation in only one RIPK1 allele in *Tank/Azi2*^DKO^/*Ripk1*^KD/+^ mice largely prevented embryonic lethality and rescued hair growth (Fig. 6a, b). Furthermore, male fertility was restored, and these mice were protected against the intravenous injection of a sublethal dose of TNF (Supplementary Fig. 6a, b). The median survival of these animals was limited to nine months, representing a significant improvement compared to *Tank/Azi2*^DKO^ mice, but not reaching the level observed in *Tank/Azi2/Tnfrsf1a*^TKO^ animals (Fig. 6c). Similarly, *Tank/Azi2*^DKO^/*Ripk1*^KD/KD^ mice, in which both RIPK1 alleles contained kinase-dead mutation, were born at a Mendelian ratio and appeared grossly normal even at nine months of age (Supplementary Fig. 6c, d), but their median survival was not further improved (Fig. 6c).

Histological analysis from five months old *Tank/Azi2*^DKO^/*Ripk1*^KD/KD^ mice revealed that the skin inflammation was fully rescued by ablation of RIPK1 kinase activity, and we observed typical deposition of subdermal fat that was not distinguishable from control *Azi2*^−/−^/*Ripk1*^KD/KD^ animals (Fig. 6d and Supplementary Fig. 6e). Staining of the kidney sections did not show signs of glomerulonephritis or amyloid deposits (Fig. 6e and Supplementary Fig. 6f). Similarly, no immune cell infiltrations were detected in samples from the pancreas, skeletal muscle, or white adipose tissue (Supplementary Fig. 6g). However, to our surprise, we noticed a strong formation of granulomas in the liver of *Tank/Azi2*^DKO^/*Ripk1*^KD/KD^ mice, accompanied by the formation of fibrotic scars (Fig. 6f and Supplementary Fig. 6h). Furthermore, we detected disrupted splenic architecture in these mice (Fig. 6g).

Biochemical analysis of the plasma from 20-24-week-old *Tank/Azi2*^DKO^/*Ripk1*^KD/KD^ mice revealed a slight but not significant increase in urea concentration. At the same time, the ionic balance was not changed compared to control *Azi2*^−/−^/*Ripk1*^KD/KD^ animals, suggesting that kidney function was not significantly impacted (Fig. 6h). Markers of liver damage, total bilirubin, and AST were increased in *Tank/Azi2*^DKO^/*Ripk1*^KD/KD^ mice as compared to control animals, although we noticed a substantial variation between animals, and the differences in AST did not reach statistical significance (Fig. 6i). Interestingly, *Tank/Azi2*^DKO^/*Ripk1*^KD/KD^ mice exhibited impaired lipid metabolism, characterized by decreased levels of plasma cholesterol, HDL, and LDL (Fig. 6j and Supplementary Fig. 6i), which accompanies liver disease in human patients and mouse models[42,43].

We observed a slight but not statistically significant decrease in albumin and an increase in total plasma protein concentration in *Tank/Azi2*^DKO^/*Ripk1*^KD/KD^ mice, which indicated only a small rise in globulin production (Fig. 6k, l). The number of splenocytes and their viability was normal in *Tank/Azi2*^DKO^/*Ripk1*^KD/KD^ mice (Supplementary Fig. 6j) and the percentage of B cells in the lymph nodes and in the spleen was not significantly different from control animals (Supplementary Fig. 6k). Interestingly, a high proportion of B cells in the spleen and lymph nodes of *Tank/Azi2*^DKO^/*Ripk1*^KD/KD^ mice exhibited an isotype-switched phenotype compared to control animals, although not to the same extent as in *Tank/Azi2*^DKO^ mice (Fig. 6m, n). Overall, our data indicate that inactivation of RIPK1 in *Tank/Azi2*^DKO^/*Ripk1*^KD/KD^ mice rescues most of the autoinflammatory phenotype observed in *Tank/Azi2*^DKO^ animals. However, these mice still develop liver inflammation, accompanied by increased activation of B cells. Therefore, these observations suggest that the TNFR1-driven autoinflammation observed in *Tank/Azi2*^DKO^ mice is primarily, but not entirely, driven by RIPK1 kinase activity.

## TANK and AZI2 are differentially recruited to TNF-RSC

Our results demonstrate that TANK and AZI2 have overlapping roles in recruiting TBK1 to TNF-RSC and protecting against TNF-induced cell death. However, although both TANK and AZI2 bind TBK1, they lack substantial sequential or structural homology[44], suggesting that they might have distinct functions in regulating TBK1 recruitment to TNF-RSC. To reveal the differences between these two adapters, we analyzed the kinetics of their association with TNF-RSC. We stimulated ST2 cells with SF-TNF for various durations and subsequently isolated TNF-RSC via Flag immunoprecipitation. TANK was strongly recruited at 15 minutes and was dissociated from the complex 30 minutes after TNF stimulation (Fig. 7a, b). The kinetics of TANK recruitment closely mirrored the recruitment of the adapter NEMO, which supports previous reports that these two proteins directly interact[45,46]. In contrast, AZI2 recruitment to TNF-RSC peaked only after 30 minutes of

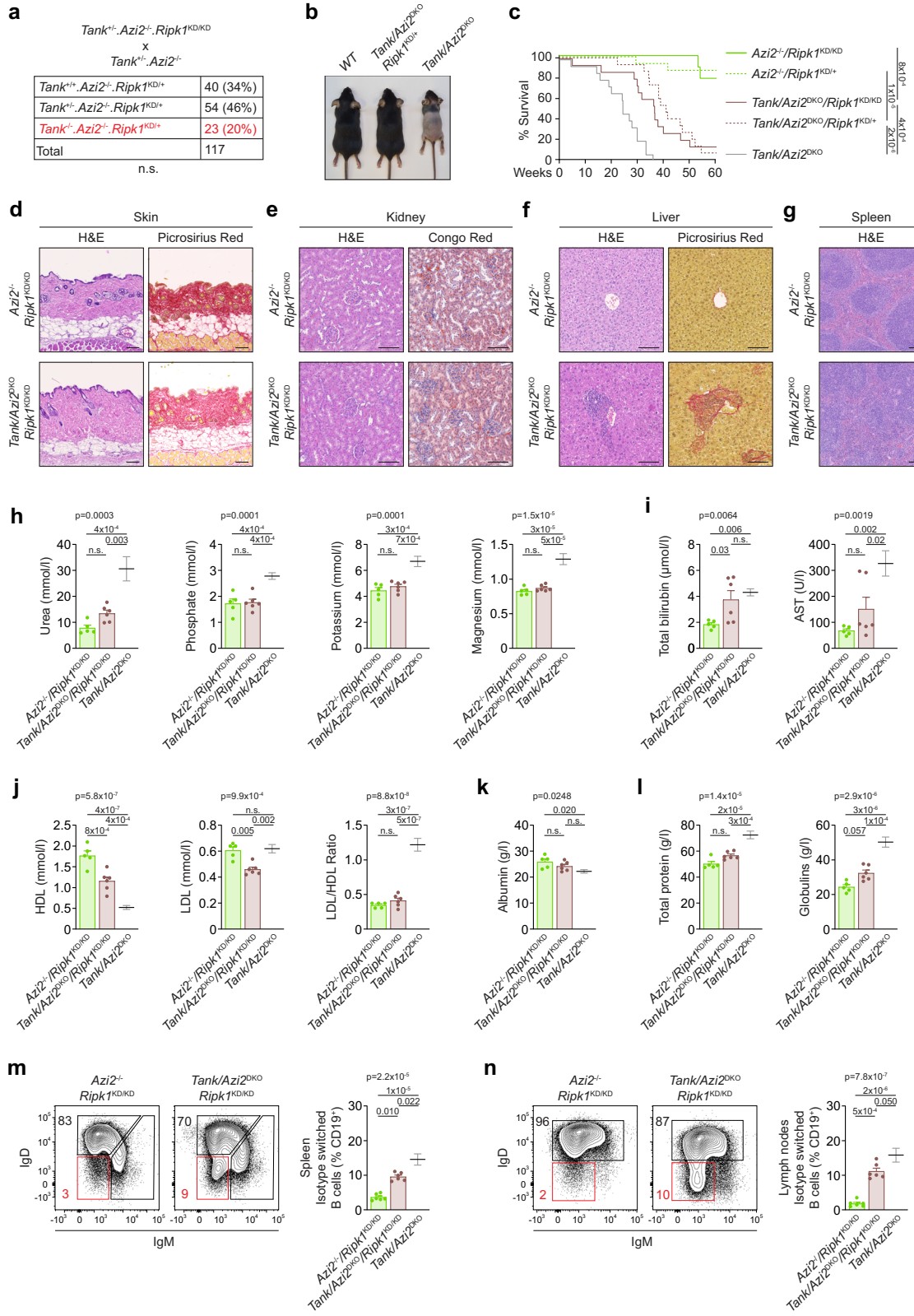

stimulation, and AZI2 subsequently remained associated with the complex. Interestingly, the recruitment kinetics of AZI2 were similar to those of the deubiquitinase A20 (Fig. 7a, b), which can remain associated with the complex for several hours due to the strong binding of A20 to M1-polyubiquitin chains while cleaving other ubiquitin linkage types[19,47,48]. The analysis of TNF-RSC isolated from MEFs and human HeLa cell lines indicated similar recruitment kinetics for TANK and

NEMO, while AZI2 recruitment was delayed and resembled that of A20 (Supplementary Fig. 7a–d).

It was previously reported that NEMO recruits both TANK and AZI2 to TNF-RSC[22]. We confirmed these findings by showing that NEMO-deficient ST2 cells had a severe defect in the TANK, AZI2, and TBK1 recruitment (Fig. 7c, d). Interestingly, our data revealed that NEMO was also required for A20 recruitment (Fig. 7c, d), consistent

**Fig. 6 | RIPK1 kinase activity promotes autoinflammation in *Tank/Azi2*$^{DKO}$ mice.** **a** Genotype and counts of pups born to parents of the indicated genotype, two-tailed chi-square test. **b** Representative photographs of 4-week-old mice of the indicated genotype. **c** Survival curves of mice of the indicated genotype (*n* = 10 for *Azi2*$^{-/-}$ and 15 for other groups). The comparison with *Tank/Azi2*$^{DKO}$ mice (based on Fig. 2) is shown in gray, log-rank Mantel-Cox test. **d**–**g** Tissue sections stained with H&E or the indicated dye. Skin (**d**), kidney (**e**), liver (**f**), or spleen (**g**) samples were collected from 20-24-week-old mice of the indicated genotype. Representative images are based on the analysis of four mice per group. **h**–**l** The concentration of different biochemical markers in the blood plasma isolated from 20-24-week-old *Azi2*$^{-/-}$/*Ripk1*$^{KD/KD}$ (*n* = 5) or *Tank/Azi2*$^{DKO}$/*Ripk1*$^{KD/KD}$ mice (*n* = 6). The comparison with *Tank/Azi2*$^{DKO}$ mice (based on Fig. 2) is shown in gray, mean + SEM, one-way ANOVA (*p*-value is indicated) with Tukey's post-tests. **m**, **n** Flow cytometry analysis of the proportion of isotype-switched B cells (CD19$^+$, IgM$^-$, IgD$^-$) in the spleen (**m**) and lymph nodes (**n**) isolated from 8-12-week-old mice of the indicated genotype (*n* = 6 per group). The comparison with *Tank/Azi2*$^{DKO}$ mice (based on Fig. 3) is shown in gray, mean + SEM, one-way ANOVA (*p*-value is indicated) with Tukey's post-tests. n.s., not significant. The scale bar is 100 μm.

with similar observations made in the context of the IL-17 receptor signaling complex[49] and Lymphotoxin β (LTβ) receptor signaling complex[50]. Furthermore, TNF-RSC isolated from the NEMO$^{KO}$ HeLa cell line did not contain TANK, AZI2, or A20 and reintroducing WT NEMO, but not NEMO lacking amino acids (AA) 211-240 required for TANK and AZI2 interaction[22,45], restored the recruitment of these proteins (Supplementary Fig. 7e, f). Therefore, NEMO is required to recruit TANK, AZI2, and also A20 to TNF-RSC.

To elucidate whether TANK and AZI2 bind directly to NEMO, we overexpressed NEMO with SF-tagged TANK or AZI2 in HEK293FT cells. Upon cell lysis, we isolated SF-TANK and SF-AZI2 via Flag immunoprecipitation. While both adapters were associated with TBK1, only SF-TANK interacted directly with NEMO and induced TBK1 phosphorylation upon overexpression (Fig. 7e, f). This suggested that AZI2 is recruited to TNF-RSC in a NEMO-dependent manner but not through direct interaction. Since AZI2 and A20 required NEMO for their association with TNF-RSC, and their recruitment kinetics were closely correlated, we aimed to explore whether A20 enables AZI2 recruitment and retention within the complex. Indeed, A20$^{KO}$ ST2 cells failed to recruit AZI2 to the TNF-RSC (Fig. 7g, h), which was rescued by the reconstitution of these cells with A20 (Supplementary Fig. 7g, h). Similarly, AZI2 recruitment was nearly abolished in HeLa cells deficient in A20 (Supplementary Fig. 7i, j).

Gene coding NEMO is located on the X chromosome, and its deficiency in males leads to prenatal lethality. However, specific mutations in this gene are tolerated, but male patients develop ectodermal dysplasia with immunodeficiency (EDA-ID). A previous study reported a patient diagnosed with EDA-ID due to L227P mutation in NEMO[51,52], which is localized in the region AA211-240 required for A20, AZI2, and TANK recruitment to TNF-RSC (Supplementary Fig. 7e, f). To test whether this mutation impacts TNF-induced TBK1 phosphorylation, we expressed NEMO WT or NEMO L227P mutant in NEMO$^{KO}$ HeLa cells. Analysis of TNF-RSC revealed that TANK, AZI2, A20, and TBK1 are absent from the complex in cells expressing NEMO L227P (Fig. 7i, j). Interestingly, NEMO L227P association with TNF-RSC was enhanced compared to NEMO WT (Fig. 7i, j), which is in accord with our observation that ablation of TANK and AZI2 promotes NEMO recruitment (Supplementary Fig. 4e, f).

Altogether, our results indicate that during the initial stage of TNF-RSC assembly, TANK directly links TBK1 with NEMO. Subsequently, NEMO facilitates A20 association with M1-ubiquitin linkages within the complex[19], which promotes the recruitment of TBK1 through AZI2 (Fig. 8a). While NEMO is released at later time points to terminate the activation of NF-κB signaling, A20 and AZI2 remain associated with the complex, ensuring the presence of TBK1 in the late stages of complex formation (Fig. 8b). In this manner, TBK1 can remain associated with the TNF-RSC to control RIPK1 activation both at the early and late stages of complex assembly.

## Discussion

In the present work, we demonstrate that the proteins TANK and AZI2 play a crucial yet redundant role in recruiting TBK1 to TNF-RSC. Cell lines deficient in TBK1 show slightly augmented activation of NF-κB and MAPK signaling pathways and are sensitive to TNF-induced cell death[20,22]. We observed similar effects in various TANK and AZI2-deficient cells, which have severe defects in TBK1 activation. Notably, although activation of the NF-κB pathway and production of

antiapoptotic proteins is enhanced in *Tank/Azi2*$^{DKO}$ cells, it is not sufficient to overcome the enhanced formation of cell death-inducing complex II. This is in accord with the notion that activation of NF-κB transcriptional responses and regulation of RIPK1 activity are two separate cell death checkpoints[23]. Although our data indicate that the major role of these two adapters is to enable the activation of TBK1 upon TNF stimulation, they may have additional functions in regulating immune homeostasis. For example, both adapters were described to interact with the kinase IKKε[53–55]. Ablation of IKKε in mice does not lead to an overt phenotype, suggesting it is not required for protection against cell death[56]. However, IKKε can partially compensate for TBK1 deficiency in specific settings, including TNF signaling[22,25].

The absence of either TANK or AZI2 in mice is well tolerated, indicating that these adapters can compensate for each other to preserve immune homeostasis in unchallenged mice. However, deficiency in both proteins triggers a severe autoinflammatory disease characterized by multiorgan inflammation, including glomerulonephritis, liver granulomas, dermatitis, and disruption of splenic architecture. Additionally, *Tank/Azi2*$^{DKO}$ animals have delayed hair growth and the absence of subdermal fat, but the precise cause of these features remains unclear. Even though the phenotype of *Tank/Azi2*$^{DKO}$ mice is very severe, it does not result in complete embryonic lethality, as observed in TBK1 knockout mice on the C57BL/6 background[24,56]. This fits with our observation that TBK1 phosphorylation was strongly decreased, but not completely blocked, in the cytoplasm of *Tank/Azi2*$^{DKO}$ cells. TBK1 is activated via autophosphorylation or phosphorylation by IKKα/β[21]. It is possible that in *Tank/Azi2*$^{DKO}$ cells, a portion of cytoplasmic TBK1 might be phosphorylated by IKKα/β and partially inhibit TNF-induced cell death.

The deletion of TNFR1 rescued *Tank/Azi2*$^{DKO}$ animals, while the ablation of RIPK1 kinase activity provided substantial, but not complete, rescue. *Tank/Azi2*$^{DKO}$/*Ripk1*$^{KD/KD}$ animals suffered from liver damage, which was accompanied by an increased proportion of isotype-switched B cells in these mice. Bone marrow transfer experiments demonstrated that *Tank/Azi2*$^{DKO}$ B cells are intrinsically prone to undergo isotype switching, which is in accordance with a previous study showing that TBK1 is a negative regulator of IgA switching in B cells[57]. Since B cells isolated from *Tank/Azi2/Tnfrsf1a*$^{TKO}$ do not show an isotype-switched phenotype, it is likely that this process relies on TNFR1 signaling but does not require RIPK1 activity.

One of our unexpected discoveries was that the absence of TANK and AZI2 leads to male sterility, which was rescued by the ablation of TNFR1 or the inactivation of RIPK1 kinase activity. This is intriguing since it shows that TNF-mediated cell death can be responsible for decreased sperm count and motility, indicating that aberrant TNF signaling might underlie fertility problems more broadly. Interestingly, TNF-induced necroptosis was suggested to promote the aging of the testes in WT mice[58]. These data open the possibility that some cases of male infertility might be caused by aberrant TNF-induced cell death and could be treated by inhibition of this pathway.

Although there is a significant functional compensation between TANK and AZI2 in vivo, they have different amino acid sequences and structures, suggesting that the existence of these two adapters is not simply due to the duplication of ancestral genes. Instead, they likely

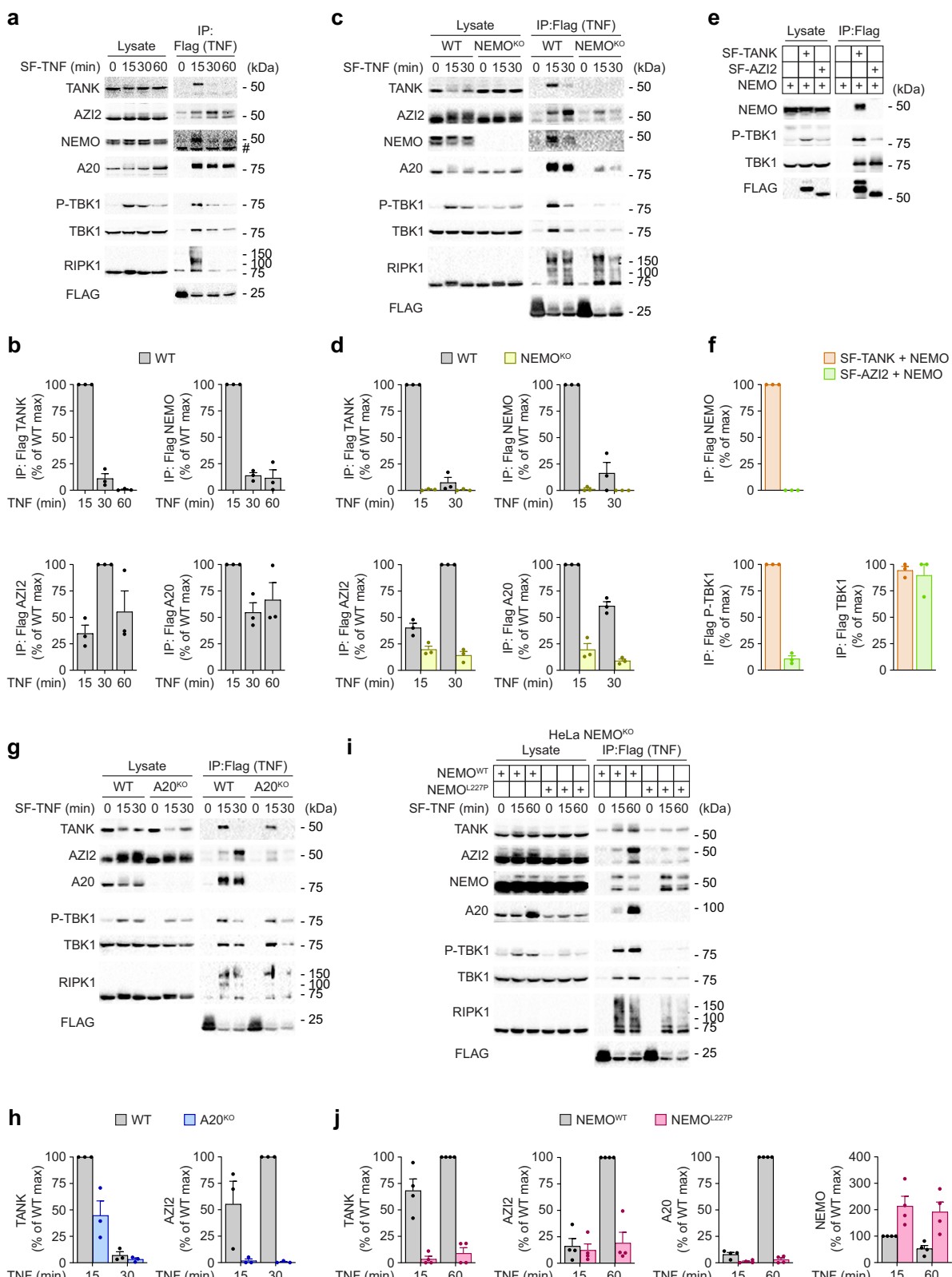

have distinct roles in controlling inflammation. Both proteins are recruited via NEMO but markedly differ in their recruitment kinetics. TANK binds directly to NEMO, which is associated with TNF-RSC during the initial stage of complex assembly. In contrast, AZI2 does not interact with NEMO but requires deubiquitinase A20, which is recruited to TNF-RSC in a NEMO-dependent manner. However, once part of the complex, A20 and AZI2 remain associated with M1-linked polyubiquitin chains even after NEMO is released[19]. Although the role of NEMO in recruiting A20 is surprising, it is in accordance with the previous study showing that short C-terminal NEMO deletion abolished recruitment of A20 to TNF-RSC and caused inflammatory disease in a human patient[59]. Our data suggest that TANK and AZI2 cooperate to enable TBK1 activation during the whole duration of TNF-RSC assembly.

**Fig. 7 | TANK and AZI2 are differentially recruited to TNF-RSC. a–d** Immunoblot analysis of TNF-RSC isolated from WT ST2 cells (**a**, **b**) or WT and NEMO[KO] ST2 cells (**c**, **d**). Cells were stimulated with SF-TNF (500 ng/ml) for the indicated time points, and lysates were subjected to anti-Flag immunoprecipitation. As a control, SF-TNF was added post-lysis to unstimulated samples. A representative experiment (**a**, **c**) and quantification of recruitment of indicated proteins to TNF-RSC from three independent experiments normalized to WT cells, mean + SEM (**b**, **d**). **e**, **f** Isolation of SF-TANK or SF-AZI2 via anti-Flag immunoprecipitation from HEK293FT cells transiently overexpressing these proteins with NEMO. A representative experiment (**e**) and quantification of three experiments normalized to WT cells, mean + SEM (**f**). **g–j** Immunoblot analysis of TNF-RSC isolated from ST2 WT or A20[KO] cells (**g**, **h**) or NEMO-deficient HeLa cells reconstituted with NEMO[WT] or NEMO[L227P] (**i**, **j**), performed as in a-d. Representative experiments (**g**, **i**) and quantification of recruitment of indicated proteins to TNF-RSC from three (**h**) or four (**j**) independent experiments normalized to the WT cells, mean + SEM. #, nonspecific band.

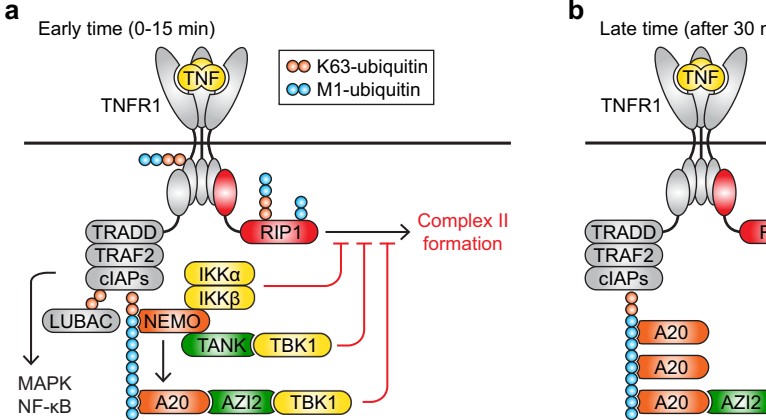

**Fig. 8 | Regulation of TBK1 activity by adapters TANK and AZI2. a**, **b** Schematic representation of the differential recruitment of TANK and AZI2 to TNF-RSC. NEMO is recruited to K63- and M1-ubiquitin linkages formed by cIAP1/2 and LUBAC, respectively. NEMO is associated with TANK-TBK1 and promotes recruitment of A20, which enables recruitment of AZI2-TBK1 (**a**). At a later time, NEMO and TANK-TBK1 are released from the TNF-RSC. However, A20 remains associated with the TNF-RSC due to M1-binding and keeps recruiting AZI2-TBK1 (**b**).

The role of TBK1 within TNF-RSC is to suppress RIPK1-mediated induction of cell death[20,22]. RIPK1 interacts directly with TNFR1 and TRADD dead domains and is recruited to TNF-RSC independently on NEMO[60–62]. The existence of two TBK1 binding adapters, one recruited early via NEMO and the second recruited later via A20, ensures that TBK1 can regulate RIPK1 activity during different stages of TNF-RSC assembly. In accord with the critical role of NEMO in recruiting TANK, A20, AZI2, and TBK1 to the TNF-RSC, we identified that human NEMO mutation L227P prevents the recruitment of all these proteins. AlphaFold prediction of NEMO structure[63] indicates that this mutation is localized in the alpha helix segment and might cause broader protein assembly disruption. Therefore, other pathogenic NEMO mutations reported in human patients might also drive the disease due to the block in TBK1 activation and enhanced TNF-induced cell death.

Combined, our results show that either TANK or AZI2 expression is sufficient to prevent the development of autoinflammatory disease. However, in the case of immune challenges, such as pathogen infection, where TNF is produced in substantial quantity for a long-term period, both adapters might be required to limit TNF-induced cell death and tissue damage. Albeit we focused on TNF signaling, it is likely that TANK and AZI2 recruit TBK1 to regulate downstream signaling from any receptor that employs NEMO and A20, such as NOD1/2, CD40, IL-1 receptor, or Toll-like receptor (TLR) family members[64]. In accord, TBK1 and TANK were reported to suppress the NF-κB signaling pathways upon stimulation of various TLRs[33,65,66]. However, whether TANK and AZI2 play a redundant role in regulating signaling from different immune receptors remains to be addressed. Collectively, our work establishes TANK and AZI2 as core components of TNF-RSC that recruit TBK1 to prevent aberrant cell death induction and subsequent development of autoinflammatory disease.

## Methods
### Cell cultures and reagents
All cell lines were cultured at 37 °C in a humidified atmosphere with 5% $CO_2$ in complete Dulbecco's Modified Eagle's Medium (DMEM) (Merck), which was supplemented with 10% fetal bovine serum (FBS) (Biosera) and 1% penicillin-streptomycin antibiotics (ATB) (Biosera). ST2 cells were kindly provided by J. Balounova, and HeLa, HEK293FT, ØNX-Eco, and ØNX-Ampho cells were kindly provided by T. Brdicka (both from the Institute of Molecular Genetics, Prague, Czech Republic). MEFs were derived from E11.5 mouse embryos and immortalized by transduction with SV40 large T antigen. To prepare BMDMs, bone marrow cells isolated from the tibia and femur of 8-9-week-old WT or *Tank/Azi2*[DKO] mice were cultured in DMEM/10% FBS/ATB supplemented with 1% non-essential amino acids (Merck) and 25 ng/ml M-CSF (Peprotech). The medium was changed after 2, 5, and 7 days. After 7 days, more than 95% of cells were positive for murine macrophage marker F4/80 as measured by flow cytometry. Subsequently, BMDMs were either directly used for experiments or cultivated with routine changing of the fully supplemented media every two days for up to 18 days. Cell lines were regularly tested for the presence of Mycoplasma using the Mycoplasmacheck services (Eurofins Genomics).

The antibodies and reagents used in this study are listed as part of Supplementary Dataset 2.

### Recombinant proteins and protein isolation
All recombinant ligands contained the CD33 leader at the N-terminus, followed by a 6xHis Tag, which enabled the secretion and purification of proteins, respectively. In the case of SF-tagged ligands, 2xStrep and 1xFlag sequences were included. The ligands contained murine TNF (AA80-235) or human TNF (AA77-233) sequences. Mouse TNF was used to stimulate ST2 cells, MEFs, and BMDMs, while human TNF was used to stimulate HeLa cells.

The coding sequences for individual ligands were prepared using the GeneArt Gene Synthesis service (Thermo Fisher Scientific) and inserted into the pcDNA3.1 vector. Plasmids were transfected into HEK293FT cells using polyethylenimine (PEI) transfection. After three days, cell supernatants containing the produced proteins were collected. Samples were loaded on a His GraviTrap TALON column

(Cytiva), previously equilibrated with a purification buffer (50 mM sodium phosphate, pH 7.4, 300 mM NaCl). The columns were washed with a purification buffer containing 20 mM imidazole and eluted with a purification buffer containing 350 mM imidazole. Subsequently, the samples were loaded on a centrifugal filter (10 kDa molecular weight cut-off, Millipore), washed several times with purification buffer to remove imidazole, and concentrated. The concentration of purified proteins was determined using NanoDrop One (Thermo Fisher Scientific). Finally, 50% glycerol was added, and for long-term storage, ligands were kept at −80 °C.

## Preparation of knockout cell lines and exogenous protein expression

ST2 and HeLa cell lines deficient in specific proteins were prepared using the CRISPR-Cas9 approach. Sequences of target sites for particular genes were selected using the CHOPCHOP web tool[67] and inserted into pSpCas9(BB)-2A-GFP (PX458) (Addgene plasmid number 48138)[68]. The target sites are listed as part of Supplementary Dataset 2. Obtained plasmids were transfected into cells using Lipofectamine 2000 (Invitrogen), and EGFP-positive cells were sorted using FACSAria™ Fusion (BD Biosciences). Single-cell clones were prepared, and the deficiency of a specific protein was verified by immunoblotting or sequencing of the DNA surrounding the target sites.

To reintroduce the proteins of interest, the coding sequences for the specified proteins were inserted into a retroviral pBabe vector, which expresses the GFP marker under the SV40 promoter. All the constructs were sequenced. Subsequently, ØNX-Eco cells (used for ST2 cells reconstitution) or ØNX-Ampho cells (used for HeLa cells reconstitution) were transfected with the pBabe-GFP vector containing coding sequences for the indicated proteins or an empty vector, using Lipofectamine 2000. Supernatants were collected, filtered through a 0.2 µl filter, and then incubated with target cells in the presence of 6 µg/ml polybrene. The cells were subsequently centrifugated (1 230 g, 30 °C, 45 min). GFP-positive cells were sorted using BD FACSAria™ Fusion.

To express proteins in HEK293FT cells, cells were transfected with NEMO and SF-TANK (SF-tag added at C-terminus) or SF-AZI2 (SF-tag at N-terminus) coding vectors using PEI.

## Analysis of signaling pathways and immunoprecipitation

ST2, MEF, BMDM, or HeLa cell lines were washed and incubated in serum-free DMEM for at least 30 minutes. Subsequently, the cells were stimulated with mouse or human TNF (100 ng/ml) for the specified time points. Next, the cells were lysed on ice in lysis buffer (30 mM Tris, pH 7.4, 120 mM NaCl, 2 mM KCl, 2 mM EDTA, 10% glycerol, 10 mM chloroacetamide, Complete protease inhibitor cocktail, and PhosSTOP tablets (Roche)) containing 1% n-Dodecyl-β-D-Maltoside (DDM) (ThermoFisher Scientific). The lysates were incubated on ice for 30 minutes and cleared by centrifugation (21 130 g, 30 minutes, 2 °C). The cleared lysates were mixed with sodium dodecyl sulfate (SDS)-containing sample buffer, reduced by 50 mM dithiothreitol (DTT), and heated at 92 °C for 3 minutes.

To isolate TNF-RSC, the cells were stimulated with SF-TNF as indicated. After solubilizing the samples in lysis buffer containing 1% DDM, the cleared lysates were subjected to immunoprecipitation with anti-Flag M2 affinity agarose gel (Merck) overnight at 4 °C. Alternatively, to isolate proteins overexpressed in HEK293FT cells, cleared lysates from cells overexpressing indicated proteins were incubated with M2 beads overnight at 4 °C. The following day, M2 beads were washed with 0.1% DDM-containing lysis buffer four times. Beads were resuspended in an SDS sample buffer containing 50 mM DTT to release immunoprecipitated proteins, and the samples were heated at 92 °C for 3 minutes.

For the immunoprecipitation of RIPK1, cells were stimulated with TNF (250 ng/ml) for 24 h in the presence or absence of the

caspase inhibitor zVAD (20 µM). RIPK1 antibody was added to pre-cleared lysates for 2 h. Subsequently, Protein A/G PLUS-Agarose (Santa Cruz Biotechnology) was added, and samples were incubated overnight at 4 °C. Upon washing the beads in lysis buffer containing 0.1% DDM, beads were resuspended in SDS sample buffer containing 50 mM DTT and heated at 92 °C for 3 minutes to elute the proteins.

The obtained samples were analyzed by immunoblotting. The chemiluminescence from HRP-fused secondary antibodies was detected using Chemidoc™ MP Imaging System and analyzed using ImageLab software v6.1 (both Bio-Rad Laboratories).

## Analysis of cell death

WT or TANK/AZI2-deficient ST2 cells, HeLa cells, or BMDMs were seeded on 96-well plates. The following day, cells were stimulated with mouse or human TNF (100 ng/ml) in the presence of 125 nM Incucyte Cytotox Red Dye (Sartorius) or 200 nM YOYO3 dye (ThermoFisher Scientific) in DMEM/10% FBS/ATB. In some experiments, IKKα/β inhibitor TPCA-1 (5 µM), caspase inhibitor zVAD (20 µM), and/or RIPK1 inhibitor Nec-1s (20 µM) were added. The induction of cell death was assessed at two-hour intervals using Incucyte Live-Cell Analysis Systems (Sartorius). Samples were analyzed in triplicates, and at least two images per well were captured.

To determine the percentage of cell death of high-throughput live cell imaging data, custom-made software was developed[69,70]. In short, brightfield- and fluorescence images (for cell death marker acquisition) of different cell lines and stages of apoptosis, necroptosis, pyroptosis, and ferroptosis induction (healthy, 30-, 60- or 90 % cell death) were acquired using Incucyte Live-Cell Analysis Systems. For necroptosis induction, HT29, HeLa cells expressing RIP3, MEFs, SCLC, and U2OS cells were treated with TNF (100 ng/ml), MCL1 inhibitor S63845 (5 µM), and zVAD (10 µM). For ferroptosis induction, HeLa, NIH-3T3, and HEK293 cells were treated with ferroptosis-inducing compound RSL3 (2 µM for NIH-3T3 and HEK293 or 4 µM for HeLa). For apoptosis induction, HeLa and SCLC cells were treated with Ccl-2 inhibitor ABT-737 (5 µM) and S63845 (5 µM). For pyroptosis induction, BMDMs were preincubated for two hours with LPS (200 ng/ml) and then treated with inflammasome inducer Nigericin (2.5 µM). To validate the specificity of cell death induction, the respective inhibitors were used in control conditions: zVAD (10 µM) for apoptosis, NSA (2 µM) or Nec-1 (10 µM) for necroptosis, Fer-1 (5 µM) for ferroptosis and DSF (5 µM) for pyroptosis. Various brightfield images were manually annotated and assembled into a data set for the training data generation. For this, the area of living and dead cells was labeled, whereby the live/dead classification was based on the presence or absence of a cell death marker. Additionally, the center of cells was labeled. This dataset was used to train a U-Net-based artificial intelligence network. This custom-made software allows the precise segmentation of cells and the prediction of whether a single cell is alive or dead using only a brightfield image as input. As output, masks of living and dead cells and their position are obtained, allowing the numerical assessment of living and dead cells. Based on those numbers, the % of cell death was calculated. Cell death prediction was validated by cell death marker.

For analysis of cell death using flow cytometry, WT or *Tank/ Azi2*^DKO ST2 cells in DMEM/10% FBS/ATB were pretreated with zVAD (20 µM) and/or Nec-1s (20 µM) for 15 minutes at 37 °C. Subsequently, TNF (250 ng/ml) was added as indicated. Cells were incubated for 24 h. After incubation, the supernatants containing dead cells were collected. The attached cells were detached with Trypsin/EDTA solution (Biosera) for 10 minutes. Subsequently, they were collected, mixed together with the supernatant, centrifugated, and resuspended in PBS containing PI (Thermo Fisher Scientific). The proportion of dead cells was determined by detecting the percentage of PI-positive cells using the BriCyte E3 flow cytometer (Mindray).

## RNA sequencing

WT and *Tank/Azi2*[DKO] BMDMs were seeded on six-well plates and stimulated with mouse TNF (100 ng/ml) for 2 h. The BMDMs were independently obtained from 3 pairs of animals. Subsequently, cells were collected, and RNA was isolated in five separate experiments using the Zymo Research Kit (R1055). The cDNA libraries were prepared using the KAPA mRNA Hyperprep Kit (Roche KK8580). Single-end sequencing was performed on Illumina NextSeq 2000 using the NextSeq 2000 P3 Reagents (50 cycles) kit (Illumina 20046810). The quality of reads was verified using FastQC v0.11.7 and MultiQC v1.17 tools[71]. The reads were aligned to mouse genome GRCm39 using salmon v1.10.0[72]. The quantification of results was done using R packages tximport v1.26.1[73]. The downstream analysis, including normalization, raw count modeling, and differential expression analyzes on complete count matrices, as well as subsets of specific samples, was done using the R package DESeq2 v1.38.3[74]. The *p*-value was calculated using the Wald test in the DESeq2 R package and corrected for multiple comparisons with the Benjamini–Hochberg procedure. The changes in gene expression are listed in Supplementary Dataset 1. Visualization and gene set enrichment analysis were performed with the packages EnhancedVolcano v1.16.0, pheatmap v1.0.12, and fgsea v1.24.0. The analysis was done on R v4.2.1.

## Mice

The mice were housed in a specific-pathogen-free facility at the Institute of Molecular Genetics of the Czech Academy of Sciences (IMG) in compliance with the laws of the Czech Republic. All animal protocols for the mice experiments were approved by the Resort Professional Commission for Approval of Projects of Experiments on Animals of the Czech Academy of Sciences, Czech Republic. The mice were provided with a standard rodent breeding diet and water ad libitum. The animal facility maintained a 12-h light and 12-h dark cycle, with a temperature of $22 \pm 1\,°C$ and a relative humidity of $55 \pm 5\%$. Experimental and control animals were co-housed, with both males and females included in the study. Experimental mice were euthanized by cervical dislocation or by isofluorane inhalation at overdose.

The murine strains utilized in this study were generated at the Czech Center for Phenogenomics at IMG. To introduce targeted indel mutations into the *Tank* and *Azi2* genes, a mixture containing Cas9 mRNA (100 ng/ml) and target-specific sgRNA (50 ng/ml) was microinjected into zygotes derived from C57BL/6 N mice. Zygote electroporation was employed to deliver sgRNAs targeting *Tnfrsf1a* and *Ripk1* genes, as described elsewhere[75]. In the case of producing the RIPK1[KD] allele harboring the inactivating D138N mutation, a single-stranded DNA template (10 μM) was included. The resulting embryos were then implanted into foster mothers.

Genomic DNA was isolated from tail biopsies of newborn mice, and the DNA sequence surrounding the CRISPR/Cas9 recognition site was amplified by PCR using Phusion polymerase (Thermo Fisher Scientific). Subsequently, the amplified DNA was sequenced. The selected founders were backcrossed onto the C57BL/6 J background for at least three generations. Genotyping primers were designed to distinguish between the WT and modified alleles using Taq DNA polymerase (TopBio). The sequences of the oligonucleotides used for generating the mice and their genotyping can be found in Supplementary Dataset 2.

## Clinical biochemistry and histology

To collect blood samples, mice were placed under terminal isoflurane anesthesia. Retro-orbital sinus bleeding was performed using glass capillaries, and the blood was collected in lithium heparin-coated tubes. After collection, each tube was gently inverted to ensure proper mixing and then kept at room temperature. The samples were centrifuged within one hour of collection (5 000 g, 10 minutes, 8 °C). The resulting plasma samples were frozen at −80 °C until further analysis. Biochemical parameters were analyzed using the AU480 clinical chemistry analyzer (Beckman Coulter). The analysis included measuring the concentrations of urea, phosphate, potassium, magnesium, total bilirubin, AST, cholesterol, triglycerides, HDL, LDL, albumin, and total protein. The concentrations of globulins were calculated as the difference between the total protein and albumin concentrations. The concentration of cytokines and antibody isotypes in the plasma was measured using the Bio-Plex 200 system (Bio-Rad). The Bio-Plex Pro Mouse Cytokine Th17 Panel A (Bio-Rad) was utilized to detect cytokines. Antibody Isotyping 7-Plex Mouse ProcartaPlex (Invitrogen) assay was employed to evaluate antibody concentrations.

Tissue samples from the back skin, kidney, liver, spleen, pancreas, skeletal muscle, white adipose tissue, and testes of 20-24-week-old mice were fixed in 4% formaldehyde solution in PBS overnight and transferred to 70% ethanol. The samples were processed by the Leica ASP6025 Vacuum Tissue Processor according to the program Standard processing overnight, embedded in Paraplast X-tra (Leica Biosystems), using the Tissue Embedding Station Leica EG1150. The five μm thick sections were prepared using the Leica Fully Motorized Rotary Microtome RM2255-FU. The obtained slides were stained by Hematoxylin H (Biognost) and Eosin Y (Carl Roth) using the Leica ST5010-CV5030 Stainer Integrated Workstation. Congo red staining was done by using the Congo red Staining Kit (Roche) in Ventana Staining Machine for Special Stains (Roche) according to producer manuals. Picrosirius red staining was done manually using Weigert's Iron Hematoxylin Set and Direct Red 80 (Merck) according to producer protocol. To detect apoptotic cells, tissue samples were stained using TUNEL Assay Kit HRP-DAB (Abcam) according to the manufacturer's instructions. Apoptotic cells with exposed 3′-OH ends of DNA fragments were marked by terminal deoxynucleotidyl transferase-mediated addition of biotinylated deoxynucleotides and detected by staining with streptavidin conjugated to horseradish peroxidase (HRP), followed by the addition of diaminobenzidine (DAB). Samples were counterstained with methyl green. Images of histological slides were obtained using the Axio Scan Z1 (Zeiss) scanner and evaluated in the software ZEN v3.4 (Zeiss). The average thickness of the subdermal fat layer was calculated by measuring 5 different locations in the skin of each analyzed animal. The proportion of swollen glomeruli was calculated from 1 mm × 1 mm areas. The number of liver granulomas was calculated from 2 mm × 2 mm areas.

## Analysis of immune cell populations

Spleen and peripheral lymph nodes were removed from 8–12-week-old mice and minced to obtain single-cell suspensions. In the case of the spleen, red blood cells were lysed in ACK buffer (150 mM $NH_4Cl$, 10 mM $KHCO_3$, 0.1 mM EDTA- $Na_2$, pH 7.4). Cells were resuspended in FACS buffer and stained on ice with LIVE/DEAD near-IR dye (Invitrogen).

To analyze B cells, the samples were stained with the following antibody panel: CD19-PE, IgM-BV421, IgD-PerCP_Cy5.5, CD23-APC, and CD1d-PE-Cy7. B cells (CD19[+]) were gated to identify isotype-switched cells (IgM[−], IgD[−]).

For the analysis of myeloid cells, the samples were stained with the following antibody panel: CD3-FITC, CD19-FITC, NK1.1-FITC, CD11b-BV421, CD11c-AF700, Ly6C-PE-Cy7, and Ly6G-AF647. Myeloid cells (CD3[−], CD19[−], NK1.1[−], CD11b[+]) were separated into three subsets: monocytes/macrophages (CD11c[−], Ly6G[−]), neutrophils (CD11c[−], Ly6G[+]), and dendritic cells (CD11c[+]). All the fluorescently labeled antibodies are listed as part of Supplementary Dataset 2.

Samples were measured on Cytek Aurora flow cytometer, and data were analyzed using FlowJo software v10.10.0 (BD Biosciences). The gating strategy is shown in Supplementary Fig. 8.

## Bone marrow transfer

Bone marrow was isolated from 4–5-week-old *Tank/Azi2*[DKO] mice harboring Ly5.2 or WT mice expressing Ly5.1/5.2. Cells were mixed in a 1:1

ratio, and $2 \times 10^6$ cells in 200 μl of PBS were intravenously injected into lethally irradiated (6 Gy) 6-week-old recipient Ly5.1 mice. After 12–14 weeks, the spleen and lymph nodes were harvested and analyzed by flow cytometry.

## Male fertility and sperm count

To evaluate male fertility, males of the specified genotype were paired with WT females for a maximum duration of 40 days or until pups were born, after which the males were separated. The number of newborn pups was recorded.

Sperm count was obtained after thorough mincing of epididymides in PBS at room temperature using a Burker chamber. Motility was determined as previously described[76] on sperm isolated by puncturing the epididymides and capacitation in Toyoda-Yokoyama-Hosi (TYH) medium at 37 °C and 5% $CO_2$ for one hour. The Computer-assisted sperm analysis system (Hamilton-Thorne, version 12.3, standard settings for mouse sperm) with Leja slides (double chamber, 0.1 mm deep) was employed to assess the percentage of motile and progressively motile spermatozoa via video recordings of over one thousand cells per male.

## Statistics and reproducibility

The used statistical tests are indicated. In the case of one-way ANOVA, two-way ANOVA, and unpaired two-tailed $t$-test, the normality was assessed using the Kolmogorov-Smirnov test or Shapiro-Wilk test. Statistical analysis was performed and graphs were plotted using Prism v10.2.3 (GraphPad).

## Reporting summary

Further information on research design is available in the Nature Portfolio Reporting Summary linked to this article.

## Data availability

The bulk RNA sequencing data have been deposited in Sequence Read Archive (SRA) under the accession code PRJNA1095284. Source data are provided with this paper.

## Code availability

The code for the custom-made software CellLocator v0.9.2 used to determine the percentage of cell death in high-throughput live cell imaging data can be accessed in the public repository Zenodo[70] [https://zenodo.org/records/13774183]. Source data are provided with this publication.

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

## Acknowledgements

We thank J. Stefanovic, L. Cupak, and H. Freislebenova for their technical assistance, D. Zudova for histopathology analysis, and Michael R. H. Vorndran for help with cell death experiments analysis. This project was supported by a Czech Science Foundation grant (21-25251S), EMBO

Installation grant (4420), and Charles University grant (PRIMUS/20/MED/003) awarded to P.D., European Union's Horizon 2020 research and innovation program under grant agreement No 802878 (ERC Starting Grant FunDiT) awarded to O.S., and Charles University grant (406322) awarded to M.P. We acknowledge Charles University institutional funding (Cooperation and UNCE/MED/016), core funding of the Institute of Molecular Genetics of the Czech Academy of Sciences, Czech Republic (IMG ASCR) (RVO 68378050), the project National Institute for Cancer Research (Program EXCELES, LX22NPO5102) and the project National Institute of virology and bacteriology (Program EXCELES, LX22NPO5103) - both funded by the European Union - Next Generation EU, and grant SVV 260637 provided by the Ministry of Education, Youth and Sports of the Czech Republic (MEYS). The project was further supported by grant LM2023036 from the MEYS provided to the Czech Center for Phenogenomics at IMG, ASCR and by the German Research Foundation (DFG) project SFB1403 (414786233) and by the German Federal Ministry of Education and Research (BMBF) project 16LW0213. A.U., M.P., and T.T. were students at the First Faculty of Medicine, and D.K., A.S., and V.N. were students at the Faculty of Science at Charles University in Prague.

## Author contributions

A.U., D.K., A.S., M.P., T.T., A.D., A.D., D.P., R.D.G., P.K., O.F. and O.M. planned, performed, and analyzed the experiments. V.N. performed RNAseq analysis. B.F.R. analyzed cell death experiments. J.P., J.L., Z.T., R.S. and O.S. supervised the experiments. P.D. conceived the study and wrote the manuscript. All authors commented on the manuscript draft.

## Competing interests

The authors declare no competing interests.
