## [Transparent Peer Review file · Nature Communications]

TBK1-associated adaptors TANK and AZI2 protect mice against TNF-induced cell death and severe autoinflammatory diseases

Corresponding Author: Dr Peter Draber

Version 0:

Reviewer comments:

Reviewer #1

(Remarks to the Author)

The paper entitled "TBK1-associated adaptors TANK and AZI2 protect against TNF-induced cell death and severe autoinflammatory disease" by Andrea Ujevic and colleagues demonstrates how deletion of both AZI2 and TANK adaptors that recruit TBK1 to the TNFRSC is required to cause spontaneous autoinflammation in mice. The manuscript builds on earlier work that identified that NEMO utilizes the adaptors TANK and AZI2 to recruit TBK1 to limit TNF-induced cell death (PMID: 304206640). Mechanistically, it now shows that NEMO recruits TANK allowing interactions with TBK1, while downstream of NEMO, A20 recruits TBK1 via AZI2 to drive a more protracted inhibitory signal for Complex II formation. Overall, the manuscript will be of high interest to the cell death field as it provides insights into how TNF receptor signaling is regulated to prevent a spectrum of disorders.

Major comments

1. A limitation of this study is that cell death and RIPK1 signaling analyses are restricted to immortalised cell lines. It will be important to demonstrate how removal of TANK and AZI2 loss in macrophages impacts TNF-driven cell death and inflammatory signaling responses, as well as cytokine production. Did the authors also consider examining B cell proliferation, isotype switching and apoptotic cell death responses in vitro to TNFR superfamily members like CD40:CD40L that engage noncanonical NF-KB but may involve TANK etc.?
2. There appears to be some discrepancy in when cell death occurs in the TANK/AZI2 cells between Fig 1D, 1E and Supplementary Figure 1A. By flow cytometry there appears to be near maximal cell death ~30% after 24 h of TNF stimulation but negligible cell death is evident by IncuCyte imaging. Please explain. Both cell death measures should also be monitored overtime for TNF plus inhibitor experiments for consistency.
3. While this study is focused on the TNF it would be interesting to define how TANK/AZI2 loss impacts TLR4-RIPK1-RIPK3 signaling in macrophages, particularly as A20, for example, can restrict RIPK3 signaling (PMID: 31086261).
4. It is shown that TANK and AZI2 loss triggers TNF-induced cell death in vitro and autoinflammation in vivo. Levels of cell death should be measured in tissues to make the connection in vivo.
5. TANK/AZI2DKO and TANK KO mice share a largely overlapping phenotype in regard to immunoglobulin responses (Fig 3). However, TANK/AZI2DKO appear to have elevated IgA, IgM and IgG1 in the plasma and an accumulation of CD19+ IgM-IgD- B cells that are suggestive of isotype switching. It will be important for the authors to better demonstrate whether autoantibody is generated to drive the multi-organ disease phenotype or if it is more associated with cell death-driven inflammation. For example, measurement of autoantibody deposition in affected tissues could be performed. Numbers of germinal centers and isotype switched antibody secreting cells/plasma cells could be quantified.
6. For clarity, please note spleen and LN weights or total cell counts plus % live cells in TANK/AZI2 DKO mice and relevant compound mutants to gauge real differences in cell populations (Fig. 3-5). Do these animals have splenic or LN enlargement?
7. Mixed bone marrow transfers (Supp Figure 2) suggest a cell intrinsic defect leads to increased isotype switched B cells in TANK/AZI2 DKO mice is complicated by an increased propensity to progenitor cell death. Are there signs that the full spectrum of diseases is transplantable if mice are reconstituted with KO bone marrow?
8. The fact the study doesn't delve into why RIPK1 kinase loss doesn't limit TNF-dependent liver and splenic pathology is a minor weakness. The authors suggest that autoantibodies may drive these pathologies without supporting this notion.

Minor

9. It appears that the levels of TBK1 may be reduced in the TANK and TANK/AZI2 double knockouts in relevant immunoblots? Please comment.
10. There is an appreciable caspase-8 p18 band in Fig. 1C in the WT ST2 cellular lysate input after 24 hours with TNF, despite no real cell death occurring in either WT or TANK/AZI2 DKO cells at this time point. Is this a timing phenomenon and, if so, wouldn't showing later time-points exaggerate any differences?
11. Please indicate in the figure legends if the TANK/AZI2 double knockout data from Figure 2 and Figure 3 is used for direct comparisons in other figures (Fig. 4 and 5) examining compound knockouts.
12. Fig 2L and Supp Fig 1H. Is there a significant difference in cholesterol or HDL in Tank^{-/-} and Azi2^{-/-}. Please modify language accordingly as it only looks like a trend.
13. Indicate in the text that the plasma cytokines were measured in aged animals with signs of hyperinflammation.
14. Does TNFR1 or RIPK1 Kinase deficiency blunt cytokine levels systemically?
15. Please check the statistical methods used for more than 2 groups and if it is appropriate.
16. The infertility issues observed in the double knockout comes out of the blue. Please consider discussion of papers that have linked RIPK3 and MLKL to testes development.

General comments

17. Line 51-52. Should state "which in turn recruit the linear ubiquitin assembly chain complex (LUBAC)"
18. Line 60. Should state "on the 129S5 background".
19. All axes representing percentages should be 0-100%.
20. Figure 1D It is difficult to interpret what relative cell death means. Please present the data as % dead cells, as done in other graphs.
21. Supplementary Figure 1A – references representative images. No images are present
22. Fig 3C Should say neutrophils. to show total cellularity of the spleen and/or spleen weights.

Reviewer #2

(Remarks to the Author)

The authors have generated double KO (DKO) mice lacking both TANK and AZI2, which serve as adaptors that link TNFR1 to TBK1. The phenotype of the DKO mice resembles that of TBK1 KO mice and can be generally suppressed by either TNFR1 or RIPK1 ablation. The results shed new light on the complex signal transduction mechanisms triggered by TNF binding to TNFR1. But how big is the advance compared to what's known already about TNFR1 signaling is a matter of debate. Moreover, as indicated below the authors seem to have missed an important component of the phenotype—a defect in NF- κ B activation.

- Fig. 1A and all other immunoblots need careful quantitation. The authors are not correct in stating that the DKO of TANK and AZI2 had no effect on NF- κ B. It is quite clear that in DKO fibroblasts, I κ B α is not induced after its degradation suggesting a defect in NF- κ B activation. This needs to be further examined by looking at p65/RelA nuclear translocation, IKK activation and RNA-seq experiments.
- Fig. 1B—What happens to other components of the TNFR1 signaling complex in DKO fibroblasts?
- Fig. 1C: I cannot see autoactivation of RIPK1 in DKO cells without TNF. Please explain. Again, no quantitation is provided. Moreover, there are no clear differences in RIPK1 phosphorylation or caspase 8 cleavage in the absence of TNF with or without ZVAD. I am not sure that the sole focus on RIPK1 is justified. After all, the inactivation of RIPK1 results in only a partial rescue (Fig. 6).
- The DKO mouse phenotypes are interesting, but it is not clear how they relate to RIPK1 activation. Furthermore, it is not clear what is the basis for increased antibody production.
- NEMO is tightly associated with IKK α/β . Thus, if NEMO recruitment is defective, IKK activation should also be defective, hence the NF- κ B activation defect.

Version 1:

Reviewer comments:

Reviewer #1

(Remarks to the Author)

This manuscript shows how the TBK-associated adaptors TANK and AZI2 protect against autoinflammatory disease driven by TNF. The authors have addressed all my concerns or justified why certain analyses were not or could not be performed. I am happy for the manuscript to be published.

Reviewer #2

(Remarks to the Author)

The revised manuscript has been improved but several important issues remain unresolved and new issues have emerged because of the revision. For instance, if TANK interacts with NEMO, how does TANK ablation affect NEMO recruitment and K63 linked ubiquitylation without causing an IKK activation defect? More issue and questions are listed below.

1. It is well established (since 1995!) that NF- κ B inhibits TNF induced apoptosis. So how comes that increased NF- κ B

activation can co-exist with enhanced cell death in Tank/Azi DKO cells ? The authors should compare the enhancement of TNF induced cell death by Tank/Azi ablation to the enhancement of TNF induced death by cycloheximide or IKKb inhibitors. I bet that NF-kB is a stronger suppressor of TNF induced cell death than Tank/Azi ablation. The authors should also use their RNA-seq analysis to compare the expression of NF-kB inducible apoptosis inhibitors (such as Bcl2 family members) in WT and Tank/Azi DKO cells (several different cell types).

2. Autoimmunity should be used to describe a pathology that is propagated by autoreactive B and T cells. Otherwise, the term autoinflammation is a more accurate descriptor of the phenotype manifested by Tank/Azi deficient mice.

3. There is also some confusion between measurements of protein kinase phosphorylation and measurements of protein kinase activity.

4. The authors seem to be reluctant to quantitate their Western blots. However, without quantitation I don't find some of the results to be entirely convincing, such as the small increase in Caspase-8 cleavage shown in Fig. 1D.

5. The absence of fat in the Tank/Azi DKO skin is not so obvious. Why not stain the sections with Oil Red O (ORO)?

6. The liver histology does not show severe liver fibrosis. It seems that the fibrosis mainly affects the bile ducts, but this needs to be examined more carefully by a liver pathologist. Bile duct fibrosis and injury would be consistent with the increase in circulating bilirubin.

7. The authors keep emphasizing the increase in IgA class switched B cells in their DKO mice without providing any mechanistic explanation. The IgA class switch is driven by TGFb, amongst other signals. They should therefore analyze TGFb signaling in different B cell subsets from their WT and DKO mice.

8. If the major function of TANK and AZI2 is to recruit and activate TBK1, which has already been established for TANK (TBK1 is Tank Binding Kinase), what is so novel about the DKO mice that we didn't already know from the analysis of Tbk^{-/-} mice?

Version 2:

Reviewer comments:

Reviewer #2

(Remarks to the Author)

I thank the authors for doing such a thorough job in addressing my remaining comments. I find the answers satisfactory and happy to recommend acceptance.

RESPONSE TO REVIEWERS' COMMENTS

Reviewer #1 (expert in signal transduction, TNF/TNFR signalling):

The paper entitled “TBK1-associated adaptors TANK and AZI2 protect against TNF-induced cell death and severe autoinflammatory disease” by Andrea Ujevic and colleagues demonstrates how deletion of both AZI2 and TANK adaptors that recruit TBK1 to the TNFRSC is required to cause spontaneous autoinflammation in mice. The manuscript builds on earlier work that identified that NEMO utilizes the adaptors TANK and AZI2 to recruit TBK1 to limit TNF-induced cell death (PMID: 30420664). Mechanistically, it now shows that NEMO recruits TANK allowing interactions with TBK1, while downstream of NEMO, A20 recruits TBK1 via AZI2 to drive a more protracted inhibitory signal for Complex II formation. Overall, the manuscript will be of high interest to the cell death field as it provides insights into how TNF receptor signaling is regulated to prevent a spectrum of disorders.

We thank the reviewer for evaluating our manuscript positively and for valuable comments on how to improve it further.

Major comments

1. A limitation of this study is that cell death and RIPK1 signaling analyses are restricted to immortalised cell lines. It will be important to demonstrate how removal of TANK and AZI2 loss in macrophages impacts TNF-driven cell death and inflammatory signaling responses, as well as cytokine production. Did the authors also consider examining B cell proliferation, isotype switching and apoptotic cell death responses in vitro to TNFR superfamily members like CD40:CD40L that engage noncanonical NF-KB but may involve TANK, etc.?

We prepared bone marrow-derived macrophages (BMDMs) from WT and *Tank/Azi2*^{DKO} animals. We evaluated the activation of signaling pathways in these cells upon TNF stimulation by immunoblotting (Fig. 4A-B) and RNA sequencing (Fig. 4C-D and Supplementary Fig. 4F). Furthermore, we studied their sensitivity to TNF-induced cell death (Fig. 4E). Our data complemented the experimental evidence showing the critical role of TANK and AZI2 in the regulation of TNF-induced responses, as described below.

Reviewer#2 prompted us to provide a detailed analysis and quantification of the activation of proinflammatory signaling pathways upon TNF stimulation in TANK/AZI2-deficient cell lines. We showed that while the absence of TANK and AZI2 leads to strong suppression of TBK1 activation, it induces mild but highly reproducible and quantifiable upregulation of NF-κB signaling pathways in a

variety of cell lines, including ST2 (Fig. 1B-C and Supplementary Fig. 1A-B), MEFs (Supplementary Fig. 4C-D), and HeLa (Supplementary Fig. 1F). These data fit with the previous report showing that TBK1-deficient cells have mildly enhanced activation of both NF- κ B and MAPK signaling pathways, likely due to the increased activation of TAK1 kinase¹. Similarly, *Tank/Azi2*^{DKO} BMDMs are unable to activate TBK1. However, compared to WT BMDMs, they exhibit a slight but very reproducible increase in TNF-induced activation of NF- κ B pathways, measured as increased phosphorylation of IKK α / β and p105, and prolonged phosphorylation and degradation of I κ B, and increased activation of JNK and p38 MAPK pathway (Fig. 4A-B).

We subsequently analyzed the TNF-induced transcriptional response of WT and *Tank/Azi2*^{DKO} BMDMs by RNA sequencing. First, we established a list of genes which is specifically upregulated in BMDMs upon TNF stimulation (Supplementary Fig. 4F). Most of these TNF-induced genes were upregulated already in unstimulated *Tank/Azi2*^{DKO} BMDMs, which is likely caused by increased basal production of TNF (Fig. 4C). The stimulation with TNF led to further enhanced expression of these genes in *Tank/Azi2*^{DKO} BMDMs as compared to WT cells (Fig. 4C-D).

The analysis of TNF-induced cell death revealed that BMDMs isolated from *Tank/Azi2*^{DKO} mice have an increased basal level of cell death, which is further markedly enhanced upon TNF stimulation compared to WT cells (Fig. 4E).

Altogether, these data provide strong evidence that the absence of TANK and AZI2 prevents TNF-induced activation of TBK1 in a variety of mouse and human cell lines and BMDMs. *Tank/Azi2*^{DKO} cells have increased cell death induction and slightly enhanced NF- κ B and MAPK signaling upon TNF stimulation, a phenotype previously reported in TBK1 deficient cells^{1,21,2}. These data further support the hypothesis that the autoinflammation observed in *Tank/Azi2*^{DKO} mice is primarily caused by defective TBK1 recruitment to TNF-RSC. We further comment on this point in the discussion.

In this manuscript, we focused primarily on TNFR1 signaling. However, it is reasonable to assume that any signaling complex employing NEMO would recruit TANK and AZI2 to enable TBK1 activation, such as CD40, IL-1R, or members of the Toll-like receptor family. In accordance with this hypothesis, we have previously discovered TANK and AZI2 in the IL-17 receptor signaling complex³. However, TBK1 might have a different and context-dependent role in regulating downstream signaling once recruited to the complex. We aim to provide a detailed analysis of the role of TANK and AZI2 in regulating other signaling complexes in follow-up studies, as it goes beyond the scope of this manuscript. We further elaborate on this issue in our response to this Reviewer's question 3. In the revised discussion section of the manuscript, we comment on the potential role of TANK and AZI2 in other signaling pathways.

2. There appears to be some discrepancy in when cell death occurs in the TANK/AZI2 cells between Fig 1D, 1E and Supplementary Figure 1A. By flow cytometry there appears to be near maximal cell death

~30% after 24 h of TNF stimulation but negligible cell death is evident by IncuCyte imaging. Please explain. Both cell death measures should also be monitored overtime for TNF plus inhibitor experiments for consistency.

In the previous version of the manuscript, we performed only a basic non-quantitative analysis of cell death measured on the Incucyte imaging system. We normalized the signal from dying cells stained with Cytotox Red Dye to the cell confluence. The result of this analysis was only a relative ‘cell death index’, which could not be interpreted as a percentage of live or dead cells.

Prompted by this Reviewer's questions 2 and 20, we reanalyzed our data using artificial intelligence-based analysis of dying cells, using previously developed algorithms⁴. This allowed us to quantify the number of live and dead cells and reveal the real-time progression of cell death as a percentage of dead cells. In the case of ST2 cells, we combined data obtained from the analysis of six independent experiments using two separate cell lines in one graph (Fig. 1E). We also provide the graph showing the effect of inhibitors zVAD and Nec-1 in blocking TNF-induced cell death measured on Incucyte. In this case, we decided to show cell death after 48 h as a bar graph which allows to visualize individual results from six independent experiments (Fig. 1F) and time-resolved analysis (Supplementary Fig. 1D). We also newly included the human cell line HeLa, in which we show that TANK and AZI2 promote TBK1 activation and their absence leads to prolonged IKK α/β activation and enhanced TNF-induced cell death (Supplementary Fig. 1F-H).

We confirmed the sensitization of *Tank/Azi2*^{DKO} ST2 cells to TNF-induced cell death via detecting propidium iodide (PI)-positive dead cells by flow cytometry. As correctly noted by the reviewer, this type of analysis led to the detection of an even higher percentage of dead cells compared to the detection of cell death using Incucyte (Supplementary Fig. 1E). In both cases, the results seem highly reproducible across experiments (as evident from comparison of Fig. 1F and Supplementary Fig. 1E). The observed differences may reflect the extensive manipulation of adherent cells required for flow cytometry analysis, as described in detail in method section. Briefly, cell supernatants containing floating dead cells are collected, and adherent live cells are subsequently trypsinized, harvested, and mixed with the floating cell fraction. Samples are centrifuged and resuspended for subsequent measurements on the flow cytometer. A plausible explanation is that the cells that would be in the process of dying due to activation of TNF-induced cell death can be further pushed towards membrane rupture by this approach, which is subsequently detected by PI staining. We comment on this nuance in the results section. Nevertheless, our data provide compelling evidence that the absence of TANK and AZI2 strongly suppresses the activation of TBK1, which is reflected in enhanced sensitivity to TNF-induced cell death.

3. While this study is focused on the TNF it would be interesting to define how TANK/AZI2 loss impacts TLR4-RIPK1-RIPK3 signaling in macrophages, particularly as A20, for example, can restrict RIPK3 signaling (PMID: 31086261).

In this project, we focused on the role of TANK and AZI2 in activating TBK1 within the TNFR1-signaling complex. The reason was that the TNF-induced cell death is the primary driver of inflammation in *Tbkl*^{KO} mice and in the *Tank/Azi2*^{DKO} mice as documented by the substantial rescue of the lethality and inflammation of these mice by crossing them to *Tnfr1*^{KO} mice (Figure 5). We fully agree with the Review that it would be highly interesting to elucidate the role of TBK1, TANK, and AZI2 in TLR4 signaling, and we are currently addressing this research question in a follow-up project, as described below.

We detected the association of TANK and AZI2 with adaptor MyD88 upon IL-1 stimulation. Besides being critical in IL-1 signaling, MyD88 is a core component of several TLR family members, including TLR4. This prompted us to analyze the role of these two adaptors in MyD88-mediated signaling. Since *Myd88* and *Azi2* are closely located on mouse chromosome 9, we employed the CRISPR/Cas9 approach to prepare *Tank/Azi2/Myd88*^{TKO} mice. Our preliminary data show that these animals are significantly protected from autoinflammatory disease and have prolonged life span. We are still collecting the experimental evidence to provide a detailed analysis of this strain and to understand the role of TANK/AZI2/TBK1 in regulating TLR and IL-1R signaling.

The analysis of this *in vivo* model requires extensive experimental work to fully understand the role of TANK/AZI2/TBK1 in regulating signaling downstream of TLR and IL-1R signaling complexes and as such it is a standalone project. We are convinced that this goes beyond the scope of the current manuscript, and we would like to publish a detailed analysis of this model in a separate study. We comment on this issue in the revised discussion section.

4. It is shown that TANK and AZI2 loss triggers TNF-induced cell death in vitro and autoinflammation in vivo. Levels of cell death should be measured in tissues to make the connection in vivo.

We thank the Reviewer for this important suggestion. We performed TUNEL staining on skin, liver, and kidney tissue sections isolated from 20-24-week WT and *Tank/Azi2*^{DKO} mice. We confirmed that the deficiency of both adaptors leads to enhanced TUNEL staining, which is in good accord with ongoing autoinflammatory disease (Supplementary Fig. 2F).

5. TANK/AZI2DKO and TANK KO mice share a largely overlapping phenotype in regard to immunoglobulin responses (Fig 3). However, TANK/AZI2DKO appear to have elevated IgA, IgM and IgG1 in the plasma and an accumulation of CD19+ IgM- IgD- B cells that are suggestive of isotype switching. It will be important for the authors to better demonstrate whether autoantibody is generated to drive the multi-organ disease phenotype or if it is more associated with cell death-driven inflammation. For example, measurement of autoantibody deposition in affected tissues could be performed. Numbers of germinal centers and isotype switched antibody secreting cells/plasma cells could be quantified.

Our data show that the production of autoantibodies by isotype-switched B cells is not the main driver of the autoinflammatory disease in *Tank/Azi2^{DKO}* mice. We propose that it is instead a phenomenon that accompanies the inflammation primarily driven by aberrant cell death. First, *Tank/Azi2^{DKO}* mice suffer from partial embryonic lethality. This phenotype is unlikely related to autoantibody production, as the adaptive immune system is not yet developed. Furthermore, most autoinflammatory phenotype, including partial embryonic lethality, is rescued in *Tank/Azi2^{DKO}/Ripk1^{KD/KD}* (Fig. 6M-N). We think that our data strongly points toward the role of TANK/AZI2 in activating TBK1 upon TNF stimulation, which is required to protect against cell death. Interestingly, the knockout of TBK1 leads to embryonic lethality. In contrast, B cell-specific deletion of TBK1 showed that this protein is crucial to restrict IgA isotype switching, but these mice do not develop severe autoinflammation as observed in *Tank/Azi2^{DKO}* mice⁵. Altogether, while it is likely that TANK and AZI2 do play a role in isotype switching, the severe phenotypic traits in these mice are directly linked to TNFR1-induced cell death.

For the histology analysis, we collected the organs without performing organ perfusion. This type of sample processing is suitable for staining for H&E or other stains used in this study. However, we cannot use these samples to analyze antibody deposits since they contain blood that has a high proportion of immunoglobulins. We would need to sacrifice a new cohort of aged mice that develop severe autoinflammatory disease to remove organs after blood perfusion to detect antibody deposits. The information provided by these experiments would probably be rather confirmatory, i.e., we would very likely observe increased antibody deposits in several organs in *Tank/Azi2^{DKO}* mice compared to WT. However, it would not provide information on whether these antibody deposits are causative of the disease or, which is much more likely, are one of many symptoms of dysregulated immune systems due to ongoing autoinflammation. We think that, in any case, the results of these experiments would lead only to an incremental knowledge gain, would not significantly improve the manuscript, and therefore would not justify additional animal experiments according to 3R principles.

6. For clarity, please note spleen and LN weights or total cell counts plus % live cells in TANK/AZI2 DKO mice and relevant compound mutants to gauge fundamental differences in cell populations (Fig. 3-5). Do these animals have splenic or LN enlargement?

We provide a representative photograph of spleens isolated from WT and *Tank/Azi2^{DKO}* mice, accompanied by graphs showing spleen weight and spleen-to-body weight ratio in these mice. Our data show that the absence of the two adaptors leads to splenomegaly (Fig. 3B). Furthermore, we provide cell count and viability for splenocytes isolated from WT, *Tank^{KO}*, *Azi2^{KO}*, and *Tank/Azi2^{DKO}* mice, which show that deficiency in the two adaptors causes increased splenic cellularity (Fig. 3D). The ablation of TNFR1 or inactivation of RIPK1 largely rescues this phenotype (Supplementary Fig. 5E and 6G)

7. Mixed bone marrow transfers (Supp Figure 2) suggest a cell intrinsic defect leads to increased isotype switched B cells in TANK/AZI2 DKO mice is complicated by an increased propensity to progenitor cell death. Are there signs that the full spectrum of diseases is transplantable if mice are reconstituted with KO bone marrow?

Our competitive experimental setup was chosen to directly compare hematopoiesis of WT and *Tank/Azi2^{DKO}* bone marrow in the same recipient mice. Indeed, in contrast to WT progenitors, bone marrow cells from *Tank/Azi2^{DKO}* mice are poorly reconstituting the hematopoiesis. We did not observe significant signs of autoinflammation in the mice reconstituted with mixed bone marrow, which would fit with our hypothesis that the autoinflammatory disease in *Tank/Azi2^{DKO}* mice is driven by aberrant TNF-induced cell death, not by hyperactivated adaptive immune system and antibody production, similarly to TBK1 knockout animals. However, it is difficult to make a strong claim based on this experiment, as there is a low number of *Tank/Azi2^{DKO}* B cells.

8. The fact the study doesn't delve into why RIPK1 kinase loss doesn't limit TNF-dependent liver and splenic pathology is a minor weakness. The authors suggest that autoantibodies may drive these pathologies without supporting this notion.

We agree that we cannot provide sufficient evidence for the role of autoantibodies in the phenotype of *Tank/Azi2^{DKO}/Ripk1^{KD/KD}* mice. Therefore, we modified the discussion to stress that although these animals have at the same time liver disease and an elevated proportion of isotype-switched B cells, it remains to be established whether this is an important driver of the remaining autoinflammatory phenotype or whether other TNFR1-mediated, but RIPK1 activity independent signaling pathways are responsible.

Minor

9. It appears that the levels of TBK1 may be reduced in the TANK and TANK/AZI2 double knockouts in relevant immunoblots? Please comment.

We performed detailed quantitative immunoblot analysis of the TBK1 expression in WT and *Tank/Azi2^{DKO}* ST2, MEFs, and BMDMs. We did not detect a significant decrease in the TBK1 protein level in *Tank/Azi2^{DKO}* cells (Fig. 4A-B and Supplementary Fig. 1A-B, 4C-D). This is indeed an important point, as it shows that the two adaptors are not required to regulate the TBK1 protein level, but instead, they function to recruit this kinase to TNF-RSC. We added a commentary on this issue in the result section of the manuscript.

10. There is an appreciable caspase-8 p18 band in Fig. 1C in the WT ST2 cellular lysate input after 24 hours with TNF, despite no real cell death occurring in either WT or TANK/AZI2 DKO cells at this time

point. Is this a timing phenomenon and, if so, wouldn't showing later time-points exaggerate any differences?

Our analysis of Incucyte measurement data shows that TNF-induced cell death of *Tank/Azi2^{DKO}* ST2 cells is detectable at 24 hours and further exacerbated at 48 hours (Fig. 1E and Supplementary Fig. 1D). Caspase-8 cleavage preceded membrane fracture and it is not unexpected that strong Caspase-8 activation is detected before the peak of cell death induction.

We also performed the analysis of Caspase-8 activation at 48 hours, but we obtained similar data showing that TNF-induced cleavage of Caspase-8 was enhanced in *Tank/Azi2^{DKO}* cells (not shown). This fits with the major role of these two adaptors in recruiting TBK1 to the TNF-RSC to protect from complex II formation and induction of cell death. We decided to show the analysis after 24 hours since it represents an earlier time point of the apoptotic pathway induction (Fig. 1D).

11. Please indicate in the figure legends if the TANK/AZI2 double knockout data from Figure 2 and Figure 3 is used for direct comparisons in other figures (Fig. 4 and 5) examining compound knockouts.

In relevant figures, we highlighted that the comparisons to data obtained from *Tank/Azi2^{DKO}* mice shown in grey are based on Figure 2, Figure 3, or Supplementary Figure 2.

12. Fig 2L and Supp Fig 1H. Is there a significant difference in cholesterol or HDL in Tank^{-/-} and Azi2^{-/-}. Please modify language accordingly as it only looks like a trend.

We modified the text since the decrease in plasma cholesterol in *Tank^{-/-}* and *Azi2^{-/-}* mice compared to WT animals does not reach statistical significance. The differences in HDL between WT and *Tank^{-/-}* or *Azi2^{-/-}* are small, but they reach statistical significance (0.0411 in the case of *Tank^{-/-}* and 0.0152 in the case of *Azi2^{-/-}*), which is now indicated in the Figure 2L. We modified the result section to reflect this observation.

13. Indicate in the text that the plasma cytokines were measured in aged animals with signs of hyperinflammation.

This is now indicated in the results section and figure legends.

14. Does TNFR1 or RIPK1 Kinase deficiency blunt cytokine levels systemically?

We could not perform this experiment because we did not have more plasma samples from our aged experimental animals, and we do not think it is worth analyzing additional cohort(s) for time, financial, and ethical reasons. The plasma levels of proinflammatory cytokines are likely decreased in *Tank/Azi2/Tnfrsf1a^{TKO}* mice and *Tank/Azi2^{DKO}/Ripk1^{KD/KD}* compared to *Tank/Azi2^{DKO}* since most of the signs of autoinflammatory disease are rescued.

15. Please check the statistical methods used for more than 2 groups and if it is appropriate.

We modified our statistical analysis. For statistical analysis of more than two groups, we employ one-way or two-way ANOVA followed by post-tests adjusted for multiple comparisons. The particular statistical test used is indicated in the figure legends.

16. The infertility issues observed in the double knockout comes out of the blue. Please consider discussion of papers that have linked RIPK3 and MLKL to testes development.

We agree with the Reviewer that rescuing male infertility in *Tank/Azi2^{DKO}* mice by removing TNFR1 is highly unexpected. However, we believe that it is fascinating because it links TNF-mediated cell death with male infertility. We added the commentary about a previous manuscript demonstrating that the activation of necroptosis accompanied by increased TNF concentration in testes contributes to the aging of the testes in WT mice⁶. This further supports the hypothesis that aberrant cell death induction in testes might lead to male sterility.

General comments

17. Line 51-52. Should state “which in turn recruit the linear ubiquitin assembly chain complex (LUBAC)”

The term LUBAC was initially proposed in the manuscript by Kirisako et al.⁷. It meant linear ubiquitin chain assembly complex. We would prefer to keep the original name for this enzymatic complex, and we also included the reference for this manuscript in the introduction.

18. Line 60. Should state “on the I29S5 background”.

We modified the text accordingly.

19. All axes representing percentages should be 0-100%.

We fully agree that the y-axis should start at 0 to avoid exaggerating small differences. This is the case in all graphs presented in this manuscript. However, it is not useful to strictly keep the y-axis upper value as 100% since it would make reading the graphs very difficult and, in some instances, rather impossible. For example, we see an increase in splenic macrophages and neutrophils while the proportion of splenic dendritic cells (DC) is decreased.

Below are the data represented in two different ways. On top, the y-axis has a variable top value, while on the bottom, the y-axis's upper value is always 100%. It is evident that the latter representation does not increase the clarity of data. On the contrary, the graphs do not allow readers to evaluate the differences between individual groups. Therefore, we prefer keeping the current visualization of graphs, which always starts at 0, but the upper limit is based on the distribution of values. We are convinced that this is a very common way of data presentation in this field and complies with journal policies and good practice in scientific publishing.

20. Figure 1D It is difficult to interpret what relative cell death means. Please present the data as % dead cells, as done in other graphs.

In the original manuscript, we analyzed our data using the basic analysis, which was not quantitative. The dead cells were detected by staining with Cytotox Red Dye (Essen Bioscience) or YOYO3 dye (ThermoFisher Scientific), and the signal was normalized to the percentage of cell culture confluence in bright-field. In the revised manuscript, we used custom software allowing us to quantify the number of live and dead cells and, based on this ratio to establish the percentage of cell death in our experiments (Fig. 1E-F, 4E, and Supplementary Fig. 1D, G-H), as detailed in the revised method section of the manuscript.

21. Supplementary Figure 1A – references representative images. No images are present

We removed these data as we combined the two knockout clones in one figure (Fig. 1E-F). Instead, we analyzed human cell lines HeLa (Supplementary Fig. 1G-H), which again show enhanced sensitivity of *Tank/Azi2^{DKO}* cells to TNF-induced cell death.

22. Fig 3C Should say neutrophils. to show total cellularity of the spleen and/or spleen weights.

We thank the Reviewer for noting the mistake in the Fig. 3C label (Fig. 3F in the revised manuscript), which was corrected. We also provide spleen cellularity and spleen weight/body weight ratio in wild-type and *Tank/Azi2^{DKO}* mice (Fig. 3B, D).

Reviewer #2 (expert in signal transduction and protein kinases):

The authors have generated double KO (DKO) mice lacking both TANK and AZI2, which serve as adaptors that link TNFR1 to TBK1. The phenotype of the DKO mice resembles that of TBK1 KO mice and can be generally suppressed by either TNFR1 or RIPK1 ablation. The results shed new light on the complex signal transduction mechanisms triggered by TNF binding to TNFR1. But how big is the advance compared to what's known already about TNFR1 signaling is a matter of debate. Moreover, as indicated below the authors seem to have missed an important component of the phenotype—a defect in NF- κ B activation.

We thank the Reviewer for prompting us to precisely define the role of TANK and AZI2 adaptors in the activation of NF- κ B. We performed detailed quantifications of our blots in a variety of cell lines to address this issue. Our data demonstrate that the ablation of the two adaptors impacts the NF- κ B pathway. However, rather than a defect in NF- κ B activation, we observed a small but highly reproducible increase in the activation of IKK α/β , which is accompanied by increased phosphorylation of I κ B and prolonged degradation of this protein. The enhanced and prolonged activation of IKK α/β explains why I κ B is not detected at later time points, as it is continuously phosphorylated and degraded. These data align with the previous publication showing that deficiency of TBK1 slightly potentiates the activation of NF- κ B and MAPK signaling pathways, likely due to the enhanced activation of TAK1¹. Our response to the Reviewer's question 1 describes the experiments related to this issue in detail.

Concerning the significance of our work, we believe that our findings establishing the combined role of TANK and AZI2 in protection against TNF-induced cell death *in vitro* and *in vivo* are important and highly relevant. Albeit it was previously suggested that TANK and AZI2 recruit TBK1 to the TNFR1 signaling complex², the functional role of these two adaptors in regulating the signaling outcome upon TNF stimulation was not analyzed. Given the major role of TBK1 in controlling TNFR1-induced signaling and cell death induction, it is important to establish how it is recruited and activated within the context of the TNFR1 receptor. Importantly, we show the role of TANK and AZI2 not only in cell lines but we document their crucial role in mouse models for the first time.

1. Fig. 1A and all other immunoblots need careful quantitation. The authors are not correct in stating that the DKO of TANK and AZI2 had no effect on NF- κ B. It is quite clear that in DKO fibroblasts, I κ B α is not induced after its degradation suggesting a defect in NF- κ B activation. This needs to be further examined by looking at p65/RelA nuclear translocation, IKK activation and RNA-seq experiments.

We carefully analyzed the immunoblots related to TNF-induced activation of signaling pathways in WT, *Tank*^{KO}, *Azi2*^{KO}, and *Tank/Azi2*^{DKO} ST2 cells (Fig. 1A in the original manuscript is Fig. 1B in the revised manuscript). Quantification from three separate experiments showed that *Tank/Azi2*^{DKO} cells have defective activation of TBK1 upon TNF stimulation. In striking contrast, the activation of IKK α/β was increased, which correlated with enhanced phosphorylation of I κ B upon TNF stimulation (Fig. 1B-

C). Phosphorylated I κ B α is rapidly degraded, and I κ B α was barely detectable 15 minutes upon TNF stimulation in WT and *Tank/Azi2*^{DKO} cells (Fig. 1B-C), indicating that *Tank/Azi2*^{DKO} cells do not have a block in NF- κ B activation. Instead, we noted that *Tank/Azi2*^{DKO} ST2 cells had a markedly prolonged phase of I κ B α degradation detected at 30 minutes upon stimulation (Fig. 1B and Supplementary Fig. 1C), which correlated with prolonged activation of IKK α / β .

To further address the effect of TANK and AZI2 deficiency on the activation of IKK α / β upon TNF stimulation, we performed additional detailed analysis of the induction of these pathways in WT and *Tank/Azi2*-deficient cells derived from ST2 (Supplementary Fig. 1A-B), MEFs (Supplementary Fig. 4C-D), and bone marrow-derived macrophages (BMDMs) (Fig. 4A-B). In all these cell lines, the absence of TANK and AZI2 led to strong suppression of TNF-induced activation of TBK1, while activation of the NF- κ B pathway was slightly but very reproducibly enhanced.

Finally, to evaluate how TANK and AZI2 impact activation of TNF-induced transcriptional response, we performed RNA sequencing analysis of WT and *Tank/Azi2*^{DKO} BMDMs stimulated or not for 2 hours with TNF. First, we identified which genes are upregulated in BMDMs upon TNF stimulation (Supplementary Fig. 4F). Interestingly, the basal expression of these genes was increased in unstimulated *Tank/Azi2*^{DKO} BMDMs, likely reflecting the increased basal expression of TNF (Fig. 4C-D). TNF stimulation of *Tank/Azi2*^{DKO} BMDMs led to further increased expression of TNF-responsive genes compared to WT cells (Fig. 4C-D).

Altogether, our data provide evidence that the absence of both adaptors led to a phenotype resembling the deficiency of TBK1, notably enhanced activation of downstream signaling pathways and cell death upon TNF stimulation. These data further support the overall conclusion of this work that the severe phenotype of *Tank/Azi2*^{DKO} mice is directly caused by the inability to recruit TBK1 to the TNF-RSC.

In the revised manuscript, we provide detailed quantification of signaling pathways triggered upon TNF stimulation of *Tank/Azi2*^{DKO} cells compared to WT cells, as discussed above. We thank the Reviewer for prompting us to perform this critical analysis. However, we think the quantification of immunoprecipitation experiments would not provide additional information, as the data are unambiguous. We believe that including quantification of these immunoprecipitation experiments would clutter the figures with unnecessary data and make the figures very large, which would affect the clarity. Therefore, we would prefer to keep the presentation of the immunoprecipitation experiments as they are.

2. Fig. 1B—What happens to other components of the TNFR1 signaling complex in DKO fibroblasts?

Prompted by our data showing increased signaling responses in *Tank/Azi2*^{DKO} cells and by the Reviewer's question number 5, we analyzed NEMO recruitment to TNF-RSC in *Tank/Azi2*^{DKO} MEFs compared to WT cells. In accordance with the enhanced activation of the NF- κ B pathway in these cells, we observed increased NEMO recruitment in *Tank/Azi2*^{DKO} MEFs (Supplementary Fig. 4E).

3. Fig. 1C: I cannot see autoactivation of RIPK1 in DKO cells without TNF. Please explain. Again, no quantitation is provided. Moreover, there are no clear differences in RIPK1 phosphorylation or caspase 8 cleavage in the absence of TNF with or without ZVAD. I am not sure that the sole focus on RIPK1 is justified. After all, the inactivation of RIPK1 results in only a partial rescue (Fig. 6).

TBK1 is crucial to prevent TNF-induced RIPK1 autophosphorylation and subsequent cell death-inducing complex II formation. However, TBK1-deficiency does not lead to spontaneous RIPK1 autoactivation in non-stimulated cells. It was published previously that unstimulated TBK1-deficient cells do not have spontaneously activated RIPK1 or cleaved Caspase-8^{1,2}. However, TNF treatment of TBK1-deficient induced strong RIPK1 activation upon TNF treatment, which led to massive cleavage of Caspase-8 and activation of the cell death pathway. Similar data were observed in cells treated with a TBK1 inhibitor^{1,2}. One notable exception is the L929 cell line, which produces autocrine TNF. Inhibition of TBK1 in this particular cell line is sufficient to induce cell death, which can be rescued by TNF-blocker Enbrel².

Similarly to *Tbk1*^{KO} cells, *Tank/Azi2*^{DKO} ST2 cells do not exhibit increased basal RIPK1 autophosphorylation, Caspase-8 cleavage, or spontaneous cell death in unstimulated cells. However, they are susceptible to TNF-induced cell death, which activates RIPK1 and leads to the induction of apoptosis. Based on these data, we concluded that these adaptors specifically suppress the formation of cell death-inducing complex II only upon TNF stimulation by recruiting TBK1 to TNF-RSC. Overall, the absence of autoactivation of RIPK1 and Caspase-8 cleavage in unstimulated *Tank/Azi2*^{DKO} cells was expected and supports our conclusions.

The fact that TBK1 prevents activation of RIPK1 upon TNF treatment but is not required to prevent spontaneous activation of RIPK1 is also evident from *in vivo* models. *Tbk1*^{KO} mice of C57BL/6 background suffer from embryonic lethality, which can be rescued by ablation of TNFR1⁸. If TBK1 were required to suppress spontaneous RIPK1 activation, the severe disease in these mice would not depend on TNF, as RIPK1 would assemble into cell death-inducing complex II even without any stimulation. Similarly, our work shows that *Tank/Azi2*^{DKO} mice suffer from partial embryonic lethality, and mice that are born suffer from very severe autoinflammatory disease. This can be rescued by genetic ablation of TNFR1. Again, if the absence of the two adaptors led to spontaneous RIPK1 autophosphorylation and subsequently to activation of apoptotic and necroptotic cell death independently on TNF stimulation, these animals would not be rescued by removing TNFR1.

We updated the manuscript to provide a better introduction to the role of TBK1 in cell death regulation, and we modified the discussion to explain how the phenotype of *Tank/Azi2*^{DKO} mice corresponds to the phenotype of *Tbk1*^{KO} mice.

4. The DKO mouse phenotypes are interesting, but it is not clear how they relate to RIPK1 activation. Furthermore, it is not clear what is the basis for increased antibody production.

Our data show that the phenotype of *Tank/Azi2*^{DKO} mice is mainly driven by TNFR1, as deletion of this protein prevents the development of the autoinflammatory disease. The inactivation of RIPK1 does not lead to complete rescue of the phenotype but is still striking. Notably, we see the rescue of partial embryonic lethality, absence of dermatitis and glomerulonephritis, and lack of immune infiltration in muscle, fat tissues, and pancreas at five months of age (Fig. 6). The newly provided photograph of 9-month-old *Tank/Azi2*^{DKO}/*Ripk1*^{KD/KD} mice further demonstrate that these mice appear normal and do not show any apparent signs of autoinflammatory disease (Supplementary Fig. 6D). In contrast *Tank/Azi2*^{DKO} develop very severe dermatitis already at five months of age (Fig. 2E). Importantly, the *Tank/Azi2*^{DKO}/*Ripk1*^{KD/KD} have significantly prolonged survival compared to *Tank/Azi2*^{DKO} mice (Fig. 6C). These data provide very compelling evidence that RIPK1 enzymatic activity is crucial in the disease progression of *Tank/Azi2*^{DKO} mice. However, it seems evident that part of the autoinflammatory phenotype observed in *Tank/Azi2*^{DKO} is mediated by TNFR1 signaling independently on RIPK1 activity. We modified the discussion to make this evident.

We agree with the Reviewer that the basis of increased antibody production in *Tank/Azi2*^{DKO} mice is not clear, albeit it seems to be a B-cell intrinsic mechanism based on bone marrow transfer experiments (Supplementary Fig. 3). Previous study analyzing mice in which TBK1 was ablated in B cells showed that TBK1 is a negative regulator of IgA switching⁵. Deletion of TANK and AZI2 may impair TBK1 function in B cells, leading to enhanced production of autoantibodies, especially IgA, which is massively increased in *Tank/Azi2*^{DKO} compared to WT mice. However, further work needs to be done specifically in B cells to address this question and we discuss this issue in the revised manuscript.

5. NEMO is tightly associated with IKK α/β . Thus, if NEMO recruitment is defective, IKK activation should also be defective, hence the NF- κ B activation defect.

As discussed above and based on detailed quantification of TNF-induced signaling pathways in *Tank/Azi2*^{DKO} and WT cells, our data show that the absence of TANK and AZI2 promote NF- κ B signaling, which is also reflected by slightly enhanced NEMO recruitment (Supplementary Fig. 4E). The differences are relatively small but highly reproducible. These data fit with a previous report showing enhanced activation of NF- κ B in *Tbkl*^{KO} cells¹, as mentioned previously.

References

- 1 Xu, D. *et al.* TBK1 Suppresses RIPK1-Driven Apoptosis and Inflammation during Development and in Aging. *Cell* **174**, 1477-1491 e1419, doi:10.1016/j.cell.2018.07.041 (2018).
- 2 Lafont, E. *et al.* TBK1 and IKKepsilon prevent TNF-induced cell death by RIPK1 phosphorylation. *Nat Cell Biol* **20**, 1389-1399, doi:10.1038/s41556-018-0229-6 (2018).
- 3 Draberova, H. *et al.* Systematic analysis of the IL-17 receptor signalosome reveals a robust regulatory feedback loop. *EMBO J* **39**, e104202, doi:10.15252/emj.2019104202 (2020).
- 4 Vorndran, M. R. H. & Roeck, B. F. Inconsistency Masks: Removing the Uncertainty from Input-Pseudo-Label Pairs. arXiv:2401.14387 (2024).
- 5 Jin, J. *et al.* The kinase TBK1 controls IgA class switching by negatively regulating noncanonical NF-kappaB signaling. *Nat Immunol* **13**, 1101-1109, doi:10.1038/ni.2423 (2012).
- 6 Li, D. *et al.* RIPK1-RIPK3-MLKL-dependent necrosis promotes the aging of mouse male reproductive system. *Elife* **6**, doi:10.7554/eLife.27692 (2017).
- 7 Kirisako, T. *et al.* A ubiquitin ligase complex assembles linear polyubiquitin chains. *EMBO J* **25**, 4877-4887, doi:10.1038/sj.emboj.7601360 (2006).
- 8 Bonnard, M. *et al.* Deficiency of T2K leads to apoptotic liver degeneration and impaired NF-kappaB-dependent gene transcription. *EMBO J* **19**, 4976-4985, doi:10.1093/emboj/19.18.4976 (2000).

RESPONSE TO REVIEWERS' COMMENTS

Reviewer #1 (Remarks to the Author):

This manuscript shows how the TBK-associated adaptors TANK and AZI2 protect against autoinflammatory disease driven by TNF. The authors have addressed all my concerns or justified why certain analyses were not or could not be performed. I am happy for the manuscript to be published.

We are glad that Reviewer 1 endorses our revised manuscript.

Reviewer #2 (Remarks to the Author):

The revised manuscript has been improved, but several important issues remain unresolved, and new issues have emerged because of the revision. For instance, if TANK interacts with NEMO, how does TANK ablation affect NEMO recruitment and K63-linked ubiquitylation without causing an IKK activation defect? More issues and questions are listed below.

This work demonstrates that adaptors TANK and AZI2 recruit TBK1 to the TNF receptor signaling complex (TNF-RSC) and enable TBK1 activation by phosphorylation. At the same time, they are not required to activate IKK α/β that initiates the NF- κ B pathway. In the following text, we recapitulate our findings describing the mechanism of TANK and AZI2 recruitment and their role in regulating TNF-induced signaling and cell death. Importantly, our data are in good accord with current literature describing that TBK1 is crucial to suppress TNF-induced cell death via preventing activation of RIPK1, not via activation of NF- κ B pathway^{1, 2}.

TANK is indeed recruited to the TNF receptor signaling complex (TNF-RSC) via direct interaction with NEMO. This is evident from several observations: (i) TANK directly binds NEMO, which was published previously^{1, 3} and we further confirmed these data (Fig. 7E-F), (ii) deficiency of NEMO prevents recruitment of TANK to TNF-RSC (Fig. 7C-D, Supplementary Fig. 7E-F), and (iii) the kinetics of TANK recruitment to TNF-RSC closely mirrors the recruitment of NEMO (Fig. 7A-D, Supplementary Fig. 7A-D).

AZI2 recruitment to TNF-RSC also requires NEMO. However, it is more complicated than in the case of TANK, since AZI2 is not directly interacting with NEMO. In this manuscript, we show that AZI2 requires A20 deubiquitinase (Fig. 7G-H, Supplementary Fig. 7G-J). A20 is recruited in the later phase of TNF-RSC assembly but remains associated with the complex for several hours due to the strong binding of A20 to M1-polyubiquitin linkages while cleaving other polyubiquitin linkage types^{4, 5, 6}. Initial A20 recruitment to TNF-RSC is mediated by NEMO, since NEMO-deficient cells cannot recruit A20 to the TNF-RSC (Fig. 7C-D and 7I-J, Supplementary Fig. 7E-F). However, once associated with the complex, A20 promotes the release of NEMO as part of the negative feedback loop, while A20 remains associated with M1-polyubiquitin linkages, which it does not cleave⁶. In accord, patients with C-terminal NEMO deletion that cannot bind A20 developed an autoimmune disease due to hyperactivated NF- κ B signaling, which was caused by the inability to recruit A20 to TNF-RSC⁷. It may

seem surprising that NEMO recruits its own negative regulator, A20. However, negative feedback loop regulation of biological reactions is relatively common, as it ensures that signaling is initiated but also properly controlled and terminated.

NEMO binds directly to M1- and K63-polyubiquitin linkages formed within the TNF-RSC^{8, 9}. Therefore, NEMO does not require TANK or AZI2 for its recruitment. This is evident from our experiments showing that NEMO mutant L227P, which is unable to recruit TANK and A20/AZI2, is still associated with the TNF receptor (Fig. 7I-J). Furthermore, TANK/AZI2^{DKO} MEFs have enhanced NEMO recruitment to TNF-RSC as compared to WT cells (Supplementary Fig. 4E-F). The reason why NEMO recruitment is even potentiated in cells lacking TANK and AZI2 is that these adaptors recruit kinase TBK1, which has two roles. First and foremost, TBK1 suppresses the activation of RIPK1-induced cell death. Second, TBK1 mildly suppresses activation of IKK α/β as part of a negative feedback loop^{1, 2}. It is important to stress that TBK1 does not block the initiation of the NF- κ B pathway upon TNF stimulation; it only regulates its magnitude and duration. We discuss this issue in more detail below in the response to the Reviewer's question 1.

The enhanced NEMO recruitment to TNF-RSC in TANK/AZI2^{DKO} cells is accompanied by increased phosphorylation of IKK α/β , increased phosphorylation of p105, and prolonged phase of I κ B α phosphorylation and degradation, which we detected in a variety of cell lines (Figure 1C-D, 4A-B, Supplementary Fig. 1A-C, 4C-D). Although the increase in activation of the NF- κ B pathway is relatively small, our analysis of several different cell lines shows that it is statistically significant (Fig. 1D, 4B and Supplementary Fig. 1B, 4D). In accord, we observed increased transcription of TNF-responsive genes in TANK/AZI2^{DKO} macrophages (Fig. 4C-D, Supplementary Fig. 4G-H). Supporting our notion that TANK suppresses activation of the NF- κ B pathway, TANK-deficient cells have enhanced NF- κ B activation upon stimulation with various TLR ligands¹⁰. Furthermore, recent publications showed that TBK1, together with related kinase IKK ϵ , suppresses IKK α/β activation upon stimulation with several members of the TLR family^{11, 12}.

Albeit NEMO is best known for activation of IKK α/β , it has more functions within TNF-RSC. We show that NEMO recruits TANK-TBK1 and also enables recruitment of A20 and subsequently AZI2-TBK1 (also depicted in the scheme in Fig. 8). Altogether, NEMO is critical for the activation of NF- κ B but also promotes recruitment of A20 and TBK1, which function as negative feedback loop regulators of NF- κ B. At the same time, as we demonstrate here, when NEMO fails to recruit TBK1 to TNF-RSC due to the absence of TANK and AZI2, it leads to RIPK1-mediated cell death and drastic autoinflammatory disease.

1. It is well established (since 1995!) that NF- κ B inhibits TNF induced apoptosis. So how comes that increased NF- κ B activation can co-exist with enhanced cell death in Tank/Azi DKO cells ?

We fully agree with the Reviewer that activation of NF- κ B is important to protect against TNF-induced cell death, and we never disputed this. However, TNF-induced cell death is checked at several

different levels. Activation of NF- κ B leads to the expression of anti-apoptotic proteins, such as Caspase-8 inhibitor cFLIP, and also induces expression of I κ B and A20, which terminate NF- κ B signaling as part of a negative feedback loop. At the same time, RIPK1, which is recruited directly to the TNF-RSC receptor, must be phosphorylated by TBK1 in order to prevent activation of RIPK1 by autophosphorylation and formation of cell-death inducing complex II^{1,2}. These cell death checkpoints function independently, but are complementary. Cells that cannot activate the NF- κ B pathway to promote transcription of anti-apoptotic genes or cells that cannot activate TBK1 to suppress RIPK1 activation are highly sensitive to TNF-induced cell death¹³. As described below, even increased activation of NF- κ B cannot compensate for TBK1 deficiency.

We have shown previously (Lafont*, Draber*, et al., Nat Cell Biol., 2018, PMID: 30420664) that the inhibition of TBK1 by a specific inhibitor MRT67307 (MRT) does not inhibit activation of NF- κ B and even slightly potentiates the activation of this pathway, measured as enhanced phosphorylation of IKK α/β and prolonged degradation of I κ B¹ (Fig. R1 below, panel A). Accordingly, a paper published independently at the same time (Xu et al., Cell, 2018, PMID: 30146158) showed that TBK1-deficient MEFs have enhanced activation of IKK α/β ² (Shown in Figure R1 below, panel B).

Figure R1. Inhibition or ablation of TBK1 slightly enhances activation of NF- κ B signalling pathway upon TNF stimulation and potentiates cell death.

A. Adapted from ¹ (Fig. 2A): A549 WT cells were pre-incubated with either vehicle (DMSO) or MRT for 30 min, followed by stimulation with TNF (200 ng/mL) for the indicated times.

B. Adapted from ² (Fig 5A): WT and Tbk1^{-/-} MEFs were stimulated with mTNF α and analyzed by immunoblotting.

However, albeit TBK1 is slightly promoting NF- κ B, its main function is to suppress the activation of RIPK1 within TNF-RSC. TBK1 phosphorylates RIPK1 to avoid its translocation to the cytoplasm and formation of cell death-inducing complex II. This control of cell death is distinct from the activation of NF- κ B transcriptional response. In accord, TBK1-deficient cells show enhanced TNF-induced cell death, which can be prevented by blocking RIPK1 activity by the specific inhibitor Necrostatin-1s (Nec-

1s)^{1,2} (Fig. R2 below, panels A and B). This is reflected by TNF-induced embryonic lethality of TBK1-deficient mice^{1,2, 14, 15}. Altogether, TBK1 is required to control TNF-induced cell death and, at the same time, negatively regulate activation of IKK α/β that induces the NF- κ B pathway. Even though TBK1^{KO} cells have elevated activation of IKK α/β and NF- κ B pathway, it is not sufficient to prevent TNF-induced cell death in TBK1-deficient cells and animals.

Figure R2. Inhibition or ablation of TBK1 potentiates TNF-induced cell death

A. Adapted from ¹ (Fig. 3B). L929 cells were treated with TNF (50 ng/mL) in the presence or absence of MRT and Nec-1s.

B. Adapted from ² (Fig 2A). MEFs were treated with 10 ng/ml mTNF α , +/or - Nec-1s (10 μ M).

In this manuscript, we demonstrated that TANK/AZI2^{DKO} cells are defective in TBK1 recruitment to TNF-RSC (Fig. 1A-B). Similarly to cells deficient in TBK1, TANK/AZI2^{DKO} cells exhibit enhanced TNF-induced phosphorylation of RIPK1 detected upon TNF/zVAD stimulation (Fig. 1E-F), increased TNF-induced cell death (Fig. 1G-H, Supplementary Fig. 1D-E, I-J) and also slightly increased NF- κ B activation upon TNF stimulation (Figure 1C-D, 4A-B, Supplementary Fig. 1A-B, 4A, 4C-D). Altogether, our data show that TANK/AZI2^{DKO} cells are sensitive to TNF-induced cell death because they cannot activate TBK1 to control RIPK1, not because they have defects in NF- κ B activation.

We modified the manuscript to stress that activation of NF- κ B transcriptional response and control of RIPK1 by TBK1 kinase are separate, albeit complementary, mechanisms of cell death protection. Therefore, the increased induction of the NF- κ B pathway in TANK/AZI2^{DKO} cells upon TNF stimulation cannot compensate for enhanced activation of RIPK1-mediated cell death caused by an inability to recruit TBK1 to TNF-RSC.

The authors should compare the enhancement of TNF induced cell death by Tank/Azi ablation to the enhancement of TNF induced death by cycloheximide or IKK β inhibitors. I bet that NF- κ B is a stronger suppressor of TNF induced cell death than Tank/Azi ablation.

As discussed above, the induction of NF- κ B and activation of TBK1 are complementary mechanisms to protect from TNF-induced cell death, and both pathways have to be activated simultaneously¹³. They function on separate levels – while activation of NF- κ B promotes transcription of anti-apoptotic genes, TBK1 directly suppresses RIPK1 activation and the formation of cell death-

inducing complex II^{1,2}. In accord, our work demonstrates that the absence of TANK/AZI2 prevents TBK1 activation, and this leads to enhanced cell death, even though NF- κ B pathway activation is slightly increased.

We newly provide experimental evidence that blocking NF- κ B activation with TPCA-1 further potentiated TNF-induced cell death in TANK/AZI2^{DKO} ST2 cells (newly added Supplementary Fig. 1F), demonstrating that IKK α/β -mediated cell death checkpoint is still operating in TANK/AZI2^{DKO} cells. Altogether, TANK and AZI2 likely prevent TNF-induced cell death by enabling TBK1 recruitment to TNF-RSC, which blocks RIPK1 activation.

The authors should also use their RNA-seq analysis to compare the expression of NF- κ B inducible apoptosis inhibitors (such as Bcl2 family members) in WT and Tank/Azi DKO cells (several different cell types).

As discussed above, TBK1 directly regulates the activity of cell death-inducing RIPK1 kinase, while it does not suppress NF- κ B activation. This was also shown previously by comparing transcriptional response in cells treated with TBK1 inhibitor MRT67307 (MRT) and IKK α/β inhibitor TPCA-1¹. While blocking IKK α/β activation massively suppressed TNF-induced transcriptional response, blocking of TBK1 did not have such an effect (Fig. R3 below).

Figure R3. TBK1 activity is not required for TNF-induced transcriptional response.

Adapted from ¹ (Fig. 2C). The heatmap illustrates the major change of expression across the dataset. The genes selected to be shown were the 100 most highly correlated with PC1.

Similarly, using RNA sequencing in this manuscript, we show that the TNF-induced transcriptional response is not suppressed in *Tank/Azi*^{DKO} macrophages, and it is even slightly increased (Fig. 4C-D).

As suggested by the Reviewer, we compared the expression of anti-apoptotic genes after two hours of TNF stimulation of WT or *Tank/Azi2*^{DKO} macrophages. At this early time point, we did not detect significantly induced transcription of anti-apoptotic Bcl2 family members genes *Bcl2*, *Bcl2l1* (BCL-XL), *Bcl2l2* (BCL-W), *Mcl1* or *Bcl2l12*. In contrast, expression of anti-apoptotic genes *Cflar* (cFLIP), *Birc2* (cIAP1), *Birc3* (cIAP2), or *Traf1* were induced upon TNF stimulation, and we detected stronger expression of these genes in *Tank/Azi2*^{DKO} macrophages (newly added Supplementary Fig. 4H).

These data further demonstrate that increased activation of TNF-induced transcriptional response in *Tank/Azi2*^{DKO} macrophages is also accompanied by increased expression of anti-apoptotic proteins. This is, however, not sufficient to suppress death, which is caused by an inability to recruit TBK1 to TNF-RSC and to directly control the activation of cell death-inducing kinase RIPK1. Altogether, our results provide very strong evidence that the cell death in TANK/AZI2^{DKO} cells is indeed caused by an inability to activate TBK1, not by a block in NF-κB signaling.

2. Autoimmunity should be used to describe a pathology that is propagated by autoreactive B and T cells. Otherwise, the term autoinflammation is a more accurate descriptor of the phenotype manifested by Tank/Azi deficient mice.

We modified the manuscript to describe the severe disease observed in *Tank/Azi2*^{DKO} mice as autoinflammation.

3. There is also some confusion between measurements of protein kinase phosphorylation and measurements of protein kinase activity.

It is established that TBK1 phosphorylation of activating serine 172 triggers TBK1 activity^{16, 17}. In accord, we stated in the manuscript that we measured TNF-induced activation of TBK1 as phosphorylation of serine 172. We have now modified the manuscript to be more precise and we changed the text to stress that we measured activating phosphorylation of TBK1.

4. The authors seem to be reluctant to quantitate their Western blots. However, without quantitation I don't find some of the results to be entirely convincing, such as the small increase in Caspase-8 cleavage shown in Fig. 1D.

We have quantified the immunoblots analyzing the activation of TNF-induced proximal signaling pathways and included careful statistical analysis (Fig. 1D, 4B, S1B, S4B, S4D) in the previous round of revisions. This demonstrated that TANK/AZI2^{DKO} cells have a block in the activation of TBK1, which is accompanied by a small but reproducible increase in phosphorylation of IKKα/β, p105, and IκB and prolonged degradation of IκB. As the differences between WT and TANK/AZI2^{DKO} cells were relatively small, a detailed analysis was necessary to make this conclusion. However, the data provided in the manuscript are now very robust in this regard.

Although quantifying immunoprecipitations experiments is not a common practice in the field, we now include quantification of all these experiments (newly added Fig. 1B, 1F, 7B, D, F, H, J and Supplementary Fig. 1H, 4F, 7B, 7D, 7F, 7H, 7J). We also quantified the analysis of complex II formation in WT and TANK/AZI2^{DKO} cells (Fig. 1F in the revised manuscript, which was previously Fig. 1D), which supports the immunoblotting data showing that TNF-induced Caspase-8 cleavage was enhanced in Tank/Azi2^{DKO} cells compared to WT cells. When cells were treated with TNF in the presence of the caspase inhibitor zVAD, which stabilizes complex II formation, we observed robust RIPK1 autophosphorylation in Tank/Azi2^{DKO} cells.

5. The absence of fat in the Tank/Azi DKO skin is not so obvious. Why not stain the sections with Oil Red O (ORO)?

Skin samples stained with H&E and picrosirius red, which stain collagens, show that subdermal fat tissue is well detectable in the skin of WT, *Tank*^{KO}, and *Azi2*^{KO} animals, while it is completely missing in *Tank/Azi2*^{DKO} mice. We observed this phenotype in all five different *Tank/Azi2*^{DKO} mice that we analyzed with two different types of staining. We now provide a graph showing the thickness of the subdermal fat layer in each mouse strain (newly added Supplementary Fig. 2F). Ablation of TNFR1 or inactivation of RIPK1 in *Tank/Azi2*^{DKO} mice rescued this phenotype (newly added Supplementary Fig. 5C, 6E).

It is not possible to stain paraffin sections analyzed in this work with the Oil Red O. This stain binds to lipids and cannot be used with the formaldehyde-fixed paraffin-embedded sections, as the alcohol used during sample preparation removes most lipids. Therefore, this control experiment would require sacrificing new cohorts of WT and *Tank/Azi2*^{DKO} mice and preparing histology samples from frozen tissue sections. Since this animal experiment would provide only confirmatory results that would not further substantiate our findings, it does not seem to be justified.

6. The liver histology does not show severe liver fibrosis. It seems that the fibrosis mainly affects the bile ducts, but this needs to be examined more carefully by a liver pathologist. Bile duct fibrosis and injury would be consistent with the increase in circulating bilirubin.

We observed granulomas throughout the liver, which impacted especially bile ducts, but not exclusively. We changed the text to reflect that we detected numerous granulomas and picrosirius red staining showed collagen-filled scars and bile duct fibrosis. To provide a better overview of the liver pathology, we provide a representative liver histology image spanning 2 mm x 2 mm. Based on these images, we constructed graphs counting the number of liver granulomas per 4 mm² in all mice analyzed in this work (newly added Supplementary Fig. 2H, 5E, 6H).

We also provide a graph showing the percentage of swollen glomeruli in the kidneys of all animal strains, which is based on the analysis of 1 x 1 mm areas (newly added Supplementary Fig. 2G, 5D, 6F).

7. The authors keep emphasizing the increase in IgA class switched B cells in their DKO mice without providing any mechanistic explanation. The IgA class switch is driven by TGF β , amongst other signals. They should therefore analyze TGF β signaling in different B cell subsets from their WT and DKO mice.

The aim of the present manuscript was to elucidate whether TANK and AZI2 are critically required for TBK1 activation and protection against TNF-induced cell death. For that purpose, we developed a new mouse model lacking both TANK and AZI2. When we characterized these mice, we noticed that they had surprisingly increased levels of plasma globulins (Fig. 2N), which was due to highly increased antibodies in the plasma, especially of IgA isotype (Fig. 3A), and it correlated with a high proportion of isotype switched B cells in these mice (Fig. 3F-I). Albeit this phenotype was interesting, it was not the driver of the autoinflammatory disease observed in *Tank/Azi2*^{DKO} mice. This is evident from the partial embryonic lethality of *Tank/Azi2*^{DKO} mice (Fig. 2C), which cannot be caused by autoantibodies, as the adaptive immune system is not yet developed. Both embryonic lethality and autoinflammation were rescued by ablation of TNFR1 (Fig. 5A-M), demonstrating that aberrant TNF signaling is the cause of autoinflammation. Furthermore, embryonic lethality was fully rescued, and autoinflammatory disease was largely prevented by inhibition of TNF-induced cell death in *Tank/Azi2*^{DKO}/*Ripk1*^{KD/KD} mice (Fig. 6A-L). Altogether, these data established that TANK and AZI2 are critical for TBK1-activation and prevention of TNF-induced cell death, mediated by RIPK1 kinase, which is the core message of our manuscript.

To elucidate the mechanism of how TANK and AZI2 regulate isotype switching is a standalone project, as it requires a very detailed analysis of B cell development and signaling and is beyond the scope of the current manuscript. Interestingly, TANK/AZI2 deficiency impacts B cell isotype switching in a cell-intrinsic manner, as demonstrated in a competitive bone marrow transfer experiment (Supplementary Fig. 3A-F). Therefore, it is likely that one or several signaling pathways important for B cell isotype switching are impacted in *Tank/Azi2*^{DKO} B cells.

Interestingly, it was previously shown that mice lacking TBK1 only in B cells had enhanced production of IgA antibodies upon immunization and in aged mice¹⁸. The TBK1-deficient B cells have moderately increased induction of NF- κ B activation upon LPS or B cell receptor stimulation (similarly as in the case of TNF-stimulation discussed above), and they have also greatly enhanced activation of non-canonical NF- κ B activation upon CD40L and BAFF stimulation. It is likely that TANK and AZI2 recruit TBK1 to any receptor that employs NEMO and A20, including CD40 or TLR family members, or to the CMB complex upon BCR signaling¹⁹. However, even if we demonstrate that the absence of TANK/AZI2 prevents TBK1 activation upon stimulation of these receptors, which is quite likely, it does not prove that this is the cause of enhanced isotype switching.

As noted by the Reviewer, TGF β signaling is another important determinant of IgA isotype switching. However, it is extremely unlikely that TANK and AZI2 would regulate canonical TGF β signaling, which signals via receptor-mediated phosphorylation of SMAD2/3²⁰. The TGF β receptor

does not activate TBK1 and also does not recruit NEMO and A20. Since, as our manuscript shows, the function of TANK and AZI2 is to connect NEMO/A20 with TBK1 (Fig. 7A-E, Supplementary Fig. 7A-E), there is no reason to assume that TANK and AZI2 proteins regulate TGF β canonical signaling. It is also unlikely that the enhanced isotype switching observed in B cells would be caused by enhanced production of TGF β cytokines because, as shown in our competitive bone marrow transfer experiments, it is a B cell-intrinsic mechanism (Supplementary Fig. 3A-F). As it is very unlikely that TANK/AZI2-deficiency would directly impact canonical TGF β signaling, raising and sacrificing new cohorts of *Tank/Azi2*^{DKO} mice to perform this experiment is not well justified.

Identifying and validating which pathway(s) regulate B cell switching in *Tank/Azi2*^{DKO} mice would require an extensive set of experiments and is beyond the scope of this manuscript that deals with the mechanism of how TBK1 is recruited to the TNFR1 complex to prevent RIPK1-mediated cell death. It is likely that TANK and AZI2 also recruit TBK1 to other signaling complexes employing NEMO and A20, such as CD40, BAFRR, or CBM complex. Since these receptors do not recruit RIPK1, the role of TANK/AZI2 and TBK1 will be different from their function in TNFR1, where they are mainly required to suppress RIPK1-mediated cell death. We modified the discussion to stress that other signaling complexes employing NEMO and A20, such as CD40, are likely to recruit TANK and AZI2 to activate TBK1.

*8. If the major function of TANK and AZI2 is to recruit and activate TBK1, which has already been established for TANK (TBK1 is Tank Binding Kinase), what is so novel about the DKO mice that we didn't already know from the analysis of *Tbk*^{-/-} mice?*

The present work aimed to establish how TBK1 is recruited to TNF-RSC to prevent RIPK1-mediated cell death. Studies of *Tbk1*^{KO} mice identified the role of TBK1 in protecting against TNF-mediated cell death and embryonic lethality. However, analysis of *Tbk1*^{KO} mice cannot provide information about the molecular mechanism enabling TNF-induced TBK1 activation.

Identifying the adaptors that recruit TBK1 to TNF-RSC is not simple or self-evident. TBK1 is a pleiotropic kinase with various functions²¹. Apart from blocking TNF-induced cell death, it is involved in regulating metabolism, activating interferon signaling, or promoting autophagy. In order to perform such a variety of functions, TBK1 is associated with various adaptors that can recruit TBK1 to different complexes. Alternatively, some receptors, such as the immune receptor STING, can interact with TBK1 directly^{22, 23, 24}. We previously proposed that AZI2 (also known as NAP1), together with TANK, might be responsible for the recruitment of TBK1 to TNF-RSC in HeLa cells¹. However, we did not pursue this further and did not test whether the combined deletion of both adaptors would have any functional effects. Whether TANK and AZI2 complement each other to promote TNF-induced activation of TBK1 and protection from cell death, both *in vitro* or *in vivo*, was not previously analyzed.

Importantly, as noted by the Reviewer and as we discuss in the introduction, it was shown that TANK is associated with TBK1, but the reported phenotype of TANK-deficient mice was relatively

mild (as shown also in our manuscript) and could have been rescued by ablation of MyD88, but not TNF. Therefore, it seemed that TANK was not required for TNF-induced TBK1 activation and protection from cell death *in vivo*¹⁰. Mice lacking AZI2 did not show any overt phenotype²⁵. It was not possible to predict whether a simultaneous deficiency in TANK and AZI2 would lead to any obvious phenotype or whether the deletion of the two proteins would be well tolerated, as in the case of mice lacking only one of these adaptors.

In this manuscript, we, for the first time, tested the hypothesis that TANK and AZI2 complement each other to promote TBK1 activation upon TNF stimulation and, by doing so, enable TBK1 to suppress RIPK1-mediated cell death. We first show this is the case of cell lines, as TANK/AZI2^{DKO} cells are sensitive to TNF-induced cell death. Subsequently, we showed that AZI2 functionally complements TANK deficiency *in vivo* to protect against severe autoinflammatory disease. Next, we demonstrated that the phenotype of *Tank/Azi2*^{DKO} mice was largely caused by TNFR1-triggered activation of RIPK1 and induction of cell death, which demonstrated the critical role of these two proteins in regulating TNF-induced cell death. Importantly, the animal models described in this work were not reported previously, and their phenotype could not have been predicted based on available data. Of note, TBK1^{KO} mice on C57BL6/N are embryonically lethal, while *Tank/Azi2*^{DKO} only exhibit partial embryonic lethality (around 50% of embryos are born), which allowed us to describe the interesting autoinflammatory phenotype in these animals.

Finally, we characterized on a molecular level how TANK and AZI2 are recruited to the TNF receptor, and we showed that they differ in their recruitment kinetics - while TANK is recruited early due to association with NEMO, AZI2 is recruited via A20 that remains associated with the receptor even after NEMO release. Again, the molecular mechanisms guiding the kinetics of TBK1 recruitment were not known previously. Our findings also indicate that it is very likely that any signaling complex employing NEMO and A20, such as CD40, would also recruit TANK and AZI2 to enable TBK1 activation. Altogether, we established the mechanism of how TBK1 is activated in the context of the TNF receptor complex to protect from cell death, both *in vitro* and *in vivo*. Understanding how TBK1 is recruited to the TNF receptor complex is important because, as our results clearly demonstrate, the inability to recruit TBK1 to the TNF receptor leads to very severe disease.

Given TNF's critical role in human health, establishing the molecular mechanism protecting against TNF-induced cell death is of wide scientific interest. We believe that this publication will be a reference point for studies analyzing the molecular mechanism of how TNF activates TBK1 to protect against RIPK1-mediated cell death.

References

1. Lafont E, *et al.* TBK1 and IKKepsilon prevent TNF-induced cell death by RIPK1 phosphorylation. *Nat Cell Biol* **20**, 1389-1399 (2018).
2. Xu D, *et al.* TBK1 Suppresses RIPK1-Driven Apoptosis and Inflammation during Development and in Aging. *Cell* **174**, 1477-1491 e1419 (2018).
3. Chariot A, Leonardi A, Muller J, Bonif M, Brown K, Siebenlist U. Association of the adaptor TANK with the I kappa B kinase (IKK) regulator NEMO connects IKK complexes with IKK epsilon and TBK1 kinases. *J Biol Chem* **277**, 37029-37036 (2002).
4. Bosanac I, *et al.* Ubiquitin binding to A20 ZnF4 is required for modulation of NF-kappaB signaling. *Mol Cell* **40**, 548-557 (2010).
5. Tokunaga F, *et al.* Specific recognition of linear polyubiquitin by A20 zinc finger 7 is involved in NF-kappaB regulation. *EMBO J* **31**, 3856-3870 (2012).
6. Draber P, *et al.* LUBAC-Recruited CYLD and A20 Regulate Gene Activation and Cell Death by Exerting Opposing Effects on Linear Ubiquitin in Signaling Complexes. *Cell Rep* **13**, 2258-2272 (2015).
7. Zilberman-Rudenko J, *et al.* Recruitment of A20 by the C-terminal domain of NEMO suppresses NF-kappaB activation and autoinflammatory disease. *Proc Natl Acad Sci U S A* **113**, 1612-1617 (2016).
8. Rahighi S, *et al.* Specific recognition of linear ubiquitin chains by NEMO is important for NF-kappaB activation. *Cell* **136**, 1098-1109 (2009).
9. Wu CJ, Conze DB, Li T, Srinivasula SM, Ashwell JD. Sensing of Lys 63-linked polyubiquitination by NEMO is a key event in NF-kappaB activation [corrected]. *Nat Cell Biol* **8**, 398-406 (2006).
10. Kawagoe T, *et al.* TANK is a negative regulator of Toll-like receptor signaling and is critical for the prevention of autoimmune nephritis. *Nat Immunol* **10**, 965-972 (2009).
11. Gitlin AD, *et al.* N4BP1 coordinates ubiquitin-dependent crosstalk within the I kappa B kinase family to limit Toll-like receptor signaling and inflammation. *Immunity* **57**, 973-986 e977 (2024).
12. Gao T, *et al.* Myeloid cell TBK1 restricts inflammatory responses. *Proc Natl Acad Sci U S A* **119**, (2022).
13. van Loo G, Bertrand MJM. Death by TNF: a road to inflammation. *Nat Rev Immunol* **23**, 289-303 (2023).
14. Bonnard M, *et al.* Deficiency of T2K leads to apoptotic liver degeneration and impaired NF-kappaB-dependent gene transcription. *EMBO J* **19**, 4976-4985 (2000).
15. Eren RO, Kaya GG, Schwarzer R, Pasparakis M. IKKepsilon and TBK1 prevent RIPK1 dependent and independent inflammation. *Nat Commun* **15**, 130 (2024).

16. Ma X, *et al.* Molecular basis of Tank-binding kinase 1 activation by transautophosphorylation. *Proc Natl Acad Sci U S A* **109**, 9378-9383 (2012).
17. Kishore N, *et al.* IKK-i and TBK-1 are enzymatically distinct from the homologous enzyme IKK-2: comparative analysis of recombinant human IKK-i, TBK-1, and IKK-2. *J Biol Chem* **277**, 13840-13847 (2002).
18. Jin J, *et al.* The kinase TBK1 controls IgA class switching by negatively regulating noncanonical NF-kappaB signaling. *Nat Immunol* **13**, 1101-1109 (2012).
19. Shimizu Y, Taraborrelli L, Walczak H. Linear ubiquitination in immunity. *Immunol Rev* **266**, 190-207 (2015).
20. Batlle E, Massague J. Transforming Growth Factor-beta Signaling in Immunity and Cancer. *Immunity* **50**, 924-940 (2019).
21. Runde AP, Mack R, S JP, Zhang J. The role of TBK1 in cancer pathogenesis and anticancer immunity. *J Exp Clin Cancer Res* **41**, 135 (2022).
22. Wild P, *et al.* Phosphorylation of the autophagy receptor optineurin restricts Salmonella growth. *Science* **333**, 228-233 (2011).
23. Fang R, *et al.* MAVS activates TBK1 and IKKepsilon through TRAFs in NEMO dependent and independent manner. *PLoS Pathog* **13**, e1006720 (2017).
24. Zhang C, Shang G, Gui X, Zhang X, Bai XC, Chen ZJ. Structural basis of STING binding with and phosphorylation by TBK1. *Nature* **567**, 394-398 (2019).
25. Fukasaka M, *et al.* Critical role of AZI2 in GM-CSF-induced dendritic cell differentiation. *J Immunol* **190**, 5702-5711 (2013).